# Reverse Forward Curriculum Learning for Extreme Sample and Demo Efficiency

**Stone Tao & Arth Shukla & Tse-kai Chan & Hao Su**
University of California, San Diego
{stao, arshukla, tsc003, haosu}@ucsd.edu

## Abstract

Reinforcement learning (RL) presents a promising framework to learn policies through environment interaction, but often requires an infeasible amount of interaction data to solve complex tasks from sparse rewards. One direction includes augmenting RL with offline data demonstrating desired tasks, but past work often require a lot of high-quality demonstration data that is difficult to obtain, especially for domains such as robotics. Our approach consists of a reverse curriculum followed by a forward curriculum. Unique to our approach compared to past work is the ability to efficiently leverage more than one demonstration via a per-demonstration reverse curriculum generated via state resets. The result of our reverse curriculum is an initial policy that performs well on a narrow initial state distribution and helps overcome difficult exploration problems. A forward curriculum is then used to accelerate the training of the initial policy to perform well on the full initial state distribution of the task and improve demonstration and sample efficiency. We show how the combination of a reverse curriculum and forward curriculum in our method, RFCL, enables significant improvements in demonstration and sample efficiency compared against various state-of-the-art learning-from-demonstration baselines, even solving previously unsolvable tasks that require high precision and control. Website with code and visualizations are here: https://reverseforward-cl.github.io/

## 1 Introduction

The RL paradigm enables a trial and error approach to learn behaviors from rewards. However, RL from scratch is sample-inefficient, especially in high-dimensional tasks with complex dynamics and/or long horizons. Even when training in fast simulators or highly GPU parallelized simulators (Makoviychuk et al., 2021; Freeman et al., 2021) a lack of priors means infeasible amounts of exploration and human programmed priors such as shaped reward functions or environment simplifications are required, none of which are remotely scalable.

Learning from demonstrations has become a popular paradigm for learning complex control skills without relying on engineered dense reward signals, complex motion planning, or infeasible amounts of compute to brute force learn solutions. However, as often is the case in robotics, demonstrations are infeasible to collect at scale, making it difficult to scale learning from demonstration methods. The classical approach to lever-

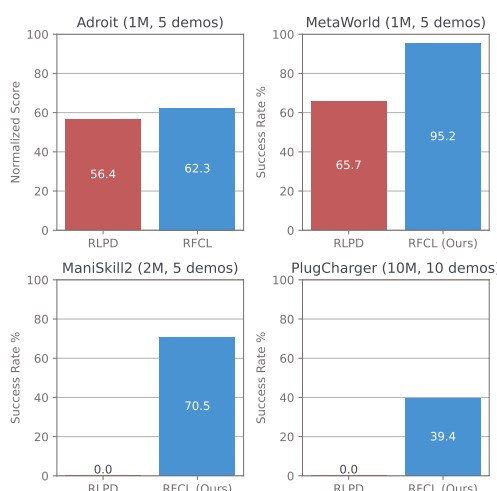

Figure 1: Results over 3 environment suites and the hardest task given fixed compute budgets ranging from 1M to 10M samples and few demonstrations. Our RFCL method drastically outperform recent approaches like JSRL and RLPD which are included in the baselines.

aging demonstrations is through behavior cloning (Atkeson & Schaal, 1997), with more recent methods performing behavior cloning via sequence modelling (Chen et al., 2021a; Zheng et al., 2022). A

recent class of methods in offline RL learn value functions from demonstrations and finetune online (Kostrikov et al., 2022; Nakamoto et al., 2023), allowing them to learn from more demonstrations including suboptimal ones.

However, many of these methods require many demonstrations even with finetuning to solve hard tasks, which raises a critical issue in domains such as robotics since robotics demonstration data is scarce. Even human demonstrations, while flexible to collect, are not only difficult to collect at scale but also exhibit undesirable properties like non-Markovianess and sub-optimality making them complicated to learn from (Mandlekar et al., 2021). Furthermore, these methods still cannot overcome the problem of exploration when it comes to sparse reward tasks, where achieving the reward in online training is infrequent, or when the task has a very wide state distribution. Such difficult tasks include long-horizon tasks (Zakka et al., 2023), highly randomized and high-precision robot manipulation tasks (Gu et al., 2023), or complex dynamics such as dexterous hand tasks (Chen et al., 2023; Rajeswaran et al., 2018).

The main motivating question then for our work is how can we effectively and practically overcome the exploration problem with access to only a few demonstrations? To address these challenges we leverage state reset from demonstrations and apply reverse and forward curriculums to online RL for sample and demonstration-efficient learning. We show how our per-demonstration reverse curriculum can overcome the exploration problems that prevent prior works from ever solving some tasks by first training an initial weak policy capable of only solving a task from a narrow initial state distribution. We then demonstrate the sample-efficiency improvements a forward curriculum brings when finetuning the reverse curriculum-trained policy to gradually learn to solve the task under the full initial state distribution.

Our primary contributions consist of our novel reverse forward curriculum learning (RFCL) algorithm and several key methods that accelerate RFCL, making it the most demonstration and sample-efficient model-free RL algorithm. We rigorously evaluate RFCL against several state-of-the-art baselines across 21 fully-observable manipulation tasks from 3 benchmarks: Adroit, ManiSkill2, and MetaWorld (Rajeswaran et al., 2018; Gu et al., 2023; Yu et al., 2019). Critically, we show that RFCL is the **only method that can solve every main task from just 5 demonstrations or less** with strong sample-efficiency in addition to fast wall-time results. Through ablations, we show how reverse and forward curriculums enable sample and demonstration-efficient learning.

## 2 RELATED WORK

**Learning from Demonstrations:** Offline collected demonstrations provide an avenue for accelerating policy learning without excessive online interactions. The classical behavior cloning approach is one method that tries to clone the exact behavior of the agent that generated the demonstrations via supervised learning (Atkeson & Schaal, 1997) and recently via sequence modelling (Chen et al., 2021a). Offline RL is another approach to imitate demonstrations without any online interactions. However, both BC and offline RL tend to be incapable of achieving high success rates/returns due to being limited to the quality and diversity of the demonstrations. Some offline RL methods address this via online finetuning (Kostrikov et al., 2022; Nakamoto et al., 2023).

Finally, there are online RL solutions that incorporate demonstration data into their objective (Rajeswaran et al., 2018; Ball et al., 2023). However, when demonstration data is extremely limited, prior approaches cannot solve complex tasks that are long-horizon, precise, and/or highly randomized due to exploration difficulty.

**Curriculum Learning:** Prior work has investigated how to incorporate curriculum learning approaches to gradually increase the difficulty of a task to facilitate training. Typically, these curriculums are hand-designed in order to solve complex long-horizon problems with sparse rewards and/or little guidance from other sources of data like demonstrations (Li et al., 2020).

Alternatively, reverse curriculums have been leveraged by initializing the agent near states that are easier to achieve meaningful rewards from, and the curriculum progresses by initializing from gradually more difficult states. Reverse curriculums can be generated via reversible dynamics (Florensa et al., 2017) or a model of backward reachability (Ivanovic et al., 2019). Some methods rollout a separate policy or demonstrator and initialize the learning agent at the end of the rollout, with some

rolling out a random number of steps (Popov et al., 2017), and recently JSRL (Uchendu et al., 2023) rolling out fewer steps over time to form a reverse curriculum.

**State Reset:** A number of approaches leverage state reset in order to overcome the problems of exploration by initializing the agent directly in more often difficult-to-reach states instead of relying on sampled actions to reach there. Some methods leverage hardcoded states or previously encountered states during interaction (Florensa et al., 2017; Ecoffet et al., 2019; Chen et al., 2023). Other methods use demonstrations as their source of initial states to initialize to, with some using a uniform curriculum over these initial states (Nair et al., 2018; Peng et al., 2018; Hosu & Rebedea, 2016), a hand-defined curriculum (Zhu et al., 2018), or a reverse curriculum (Resnick et al., 2018; Salimans & Chen, 2018).

Our method is similar to some prior work in that we leverage a reverse curriculum and reset to states in demonstrations. Unique to our approach, however, is the ability to efficiently leverage more than one demonstration via a per-demonstration reverse curriculum, different to Resnick et al. (2018) which only uses one demonstration and tackles tasks with small initial state distrbutions, and different to Nair et al. (2018) which uses multiple demonstrations but applies a uniform curriculum and struggles with more difficult tasks due to exploration inefficiency. Furthermore, we propose a novel approach of coupling reverse curriculum with a forward curriculum, which enables greater demo efficiency and drastically improves sample efficiency compared to state-of-the-art baselines. Moreover, we introduce key methods that further accelerate learning in our curriculum designs not done by past work. These contributions enable our approach to even solve some previously unsolved environments from sparse rewards that even human designed dense rewards are insufficient for.

## 3 PRELIMINARIES

### 3.1 PROBLEM SETTING

We consider the standard Markov Decision Process (MDP) which can be described as a tuple $\mathcal{M} = (\mathcal{S}, \mathcal{A}, \mathcal{R}, \mathcal{T}, \rho, \gamma)$ where $\mathcal{S}$ is the continuous state space, $\mathcal{A}$ is the continuous action space, $\mathcal{R} : \mathcal{S} \times \mathcal{A} \to \mathbb{R}$ is the scalar reward function, $\mathcal{T} : \mathcal{S} \times \mathcal{A} \to \mathcal{S}$ is the environment dynamics function, $\rho$ is the initial state distribution, and $\gamma \in [0, 1]$ is the discount factor. The goal is to learn a policy $\pi_\theta : \mathcal{S} \to \mathcal{A}$ parameterized by $\theta$ that maximizes the expected discounted return, namely solving $\max_\theta E_{\pi_\theta}[\sum_{t=0}^{\infty} \gamma^t r_t]$ where $r_t$ is the reward at timestep $t$. One goal in RL is sample efficiency, learning a $\pi_\theta$ that maximizes the discounted return with as few environment interactions as possible.

In this work, we consider complex MDPs with sparse reward functions where +1 is given for being in a success state and 0 otherwise. This setting is motivated by how many environments in embodied AI research such as robotics are high-dimensional with difficult dynamics. Moreover, dense reward functions are non-trivial to construct and may be sub-optimal (Singh et al., 2019; Amodei et al., 2016). While sparse reward functions are desirable as they reflect directly our desired goal (e.g. high success rates), they are more difficult to learn from due to exploration. A common approach is to leverage pre-collected demonstration data to aid in overcoming the problems of exploration in sample-efficient learning under sparse rewards. We define a dataset of demonstrations $D = \{\tau_0, \tau_1, ..., \tau_N\}$ to be a set of $N$ demonstrations where $\tau_i = (s_{i,0}, a_{i,0}, ..., s_{i,T_i-1}, a_{i,T_i-1}, s_{i,T_i})$ is a trajectory of length $T_i$ composed of a sequence of states and actions. In practice, the observations used by the policy $\pi_\theta$ may be different from the actual environment state but for simplicity in this paper state also refers to observation.

### 3.2 CURRICULUM LEARNING

In RL, curriculum learning is an approach where during training, the agent is presented with gradually harder tasks, each leveraging skills from the previous easier tasks in order to accelerate learning, mimicking how humans learn (Narvekar et al., 2020). In this paper, the MDP $\mathcal{M}$ is changed throughout training by modifying the initial state distribution $\rho$ to facilitate a curriculum. Define $\mathcal{S}_{\text{init}}$ to be the original initial state distribution of the MDP we wish to maximize discounted return. We can then define a curriculum as a sequence of initial state distributions $(\rho_0, \rho_1, ...)$; each of the elements are also referred to as stages of the curriculum. Each curriculum also comes with a criterion for whether to advance to the next stage of the curriculum from $\rho_i$ to $\rho_{i+1}$.

A key novelty of our work is leveraging both **reverse curriculums and forward curriculums**. The reverse curriculum initial state distributions are denoted with $\rho^r$ and the forward curriculum with $\rho^f$. In the reverse curriculum, we reset to states in the demonstrations similar to Nair et al. (2018). Different to prior work we auto generate a per-demonstration reverse curriculum around these demonstration states in order to accelerate learning under sparse rewards. The reverse curriculum enables sample-efficient learning of solving the MDP from the narrow set of initial states in the demonstrations $\{s_{i,0}\}$, detailed in Sec. 4.1. In the forward curriculum, we adopt a similar approach to Prioritized Level Replay (PLR) (Jiang et al., 2021) which assumes no priors about the parameterization of the state. Different to the reverse curriculum, we reset to states sampled from $\mathcal{S}_{\text{init}}$ instead of states in demonstrations. The curriculum here is constructed in a way that prioritizes initial states that are at the edge of the ability of the policy $\pi_\theta$. By sampling these initial states more frequently, the training focuses on learning to solve from initial states just within reach of the agent, and as the agent improves, the curriculum advances to provide more difficult initial states, enabling the agent to solve the majority of initial states in $\mathcal{S}_{\text{init}}$. This is detailed in Sec. 4.2, and results in our demonstration ablations in Sec. 5.2 demonstrate how the forward curriculum coupled with the reverse curriculum enables our algorithm to use fewer demonstrations than just using a reverse curriculum and significantly fewer demonstrations than prior work.

## 4    REVERSE FORWARD CURRICULUM LEARNING (RFCL)

The goal of our method is to provide a practical and flexible algorithm that accelerates learning in complex environments with a limited number of demonstrations without shaped rewards. A visual overview of the two stage training process of RFCL is in Fig. 2. In stage 1, a reverse curriculum is generated via state resets to states in demonstrations, learning a policy to initially solve a narrow initial state distribution. The reverse curriculum and state resets help overcome difficult exploration problems by initializing the agent near success states and gradually moving it further away. The forward curriculum then generalizes the initial policy to

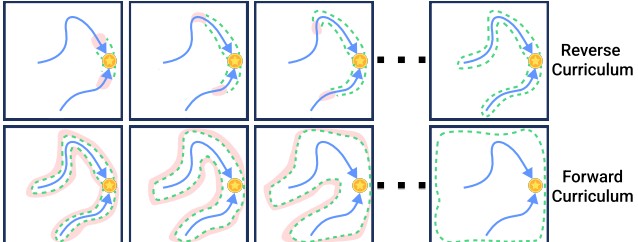

Figure 2: A simplfied view of the reverse and forward curriculum. The blue arrows represent the given demonstration trajectories (2 in this example), starting from an initial state and moving towards the goal marked by a gold star. The area covered by dashed green lines represent the distribution of initial states from which the policy can achieve high return. The area shaded in red represents the most frequently sampled initial states during each stage of curriculum. From left to right represents the progression of the trained policies ability over the course of the curriculum training.

a larger initial state distribution without using more demonstration data by gradually sampling more difficult initial states to train on over time, enabling additional demonstration and sample efficiency. In both stages, we use the off-policy algorithm Soft Actor Critic (Haarnoja et al., 2018) with a Q-ensemble (Chen et al., 2021b), see hyperparameters in Appendix C.

### 4.1    STAGE 1: REVERSE CURRICULUM

In this stage, we start with randomly initialized actor and critic networks and train using standard RL with SAC. We further aggressively oversample a separate offline buffer consisting of all the given demonstration data, optimal and sub-optimal, as done in Hansen et al. (2023); Ball et al. (2023) during online RL by sampling 50% of data from the online buffer and the rest from the offline buffer during optimization. We further adopt a per-demonstration reverse curriculum that improves sample-efficiency compared to prior work that use other kinds of reverse curriculums or demonstration state resets. Finally, we introduce a few key methods that further accelerate reverse curriculum learning.

**Per-demonstration Reverse Curriculum Construction:** We observe that not all demonstrations are similar due to different initial states and are often multi-modal, as is the case of human or motion-planned demonstrations. As a result, a curriculum for each demonstration is necessary as opposed to a curriculum constructed from all demonstrations as done in prior work. A per-demo

curriculum ensures noisy information arising from the multi-modality of demonstrations do not impact the reverse curriculum of each demonstration as much.

For each successful demonstration $\tau_i$ we assign a start step $t_i$. For each new episode during training, we first sample a demonstration $\tau_i$ from our dataset $D$ with probability $t_i/T_i$, sampling demonstrations progressing slower more frequently. Then, we sample a discrete offset value $k \sim K$ and reset the environment to the state $s_{i,t+k}$ from $\tau_i$, the state at the $t+k$ timestep of demonstration $\tau_i$. We make the design choice of using a geometric distribution as $K$.

Initializing all $t_i = T_i$, the horizon of $\tau_i$, at the start of training, we train mostly on states that are success states as these demonstrations are successful. As a result, it is highly likely to achieve positive rewards during training and thus easy to learn. We also define a small reverse step size $\delta$ constituting the distance between stages in the curriculum. Thus, the reverse curriculum for each demonstration $\tau_i$ is defined as $(\rho_{T_i}^r, \rho_{T_i-\delta}^r, ..., \rho_0^r)$ where $\rho_t^r$ samples $s_{i,t+k}$ with probability $p_K(k)$.

We now define the stage transition criterion. During training and at curriculum stage $\rho_t^r$, whenever we sample $s_{i,t}$ (when $k = 0$), we are sampling an initial state that is at the frontier of the agent's abilities. If the last $m$ times the agent achieved success at the end of episodes initialized at state $s_{i,t}$, then we progress the curriculum of demonstration $\tau_i$ to the next stage $\rho_{t-\delta}^r$. This ensures the agent only begins to tackle the task from the slightly more difficult demonstration state $s_{i,t-\delta}$ provided it is already succesful enough on $s_{i,t}$, forming the curriculum.

Once all demonstration curriculums are at stage $\rho_0^r$, the reverse curriculum is considered complete, as now the agent can achieve a high return on all demonstration initial states $s_{i,0}$. We visualize how the reverse curriculum progresses in Fig. 2 and via videos on the project page. The red area demonstrates how the sampled initial states slowly reverse their way back to the first states of the two demonstrations, training the policy in reverse to perform well on the narrow initial state distribution visualized by the green region. An empirical investigation via a simple pointmaze is done as well in Sec. 5.2. We benchmark alternative curriculum choices in addition and show the sample efficiency gains of the per-demonstration reverse curriculum in Table 1.

**Dynamic Episode Timelimits:** Since we are often resetting to states not from the true initial state distribution $\mathcal{S}_{\text{init}}$ but from various points in demonstrations, the time it takes to achieve success from these states as quickly as possible varies significantly. Observe that if we sample a state near a success state that yields positive reward, we need far less environment interactions to then get the positive reward compared to sampling a state farther away. This motivates the use of a dynamic timelimit depending on which state is sampled. Suppose we sample state $s_{i,t}$, then a simple choice is to set the episode timelimit to $1 + (T_i - t)\phi^{-1}$. $\phi$ is hyperparameter for the ratio of demonstration length to episode horizon. We fix the value of $\phi$ for each environment suite and generally for better sample-efficiency $\phi$ can be set larger if the source demonstration is fairly slow (e.g. human demonstrations) and smaller if the source demonstration is fast or optimal (e.g. scripted policies). We show an ablation on dynamic epsiode timelimits on stage 1 training and demonstrate the improved sample efficiency in Table 1.

## 4.2 STAGE 2: FORWARD CURRICULUM

In this stage, we continue the training of the actor and critics from stage 1 without resetting any networks. We instead reset the online buffer and initialize a new offline buffer. Like stage 1, we continue to aggressively sample from a separate offline buffer. Different this time is that we fill the offline buffer with the online buffer of the stage 1 training, with the motivation being to ensure the value functions do not suddenly have to learn from completely unseen data and avoid unlearning phenomenon encountered by past work investigating finetuning of pretrained policies (Peng et al., 2019; Kostrikov et al., 2022). Moreover, our choice of RL algorithm is capable of learning from sub-optimal data so it is helpful to add in the data that was collected from stage 1 training instead of just using successful demonstrations. Crucial to this stage is the forward curriculum described below, which enables more demonstration-efficient learning.

**Forward Curriculum Construction:** Following the reverse curriculum stage, we will have trained a policy $\pi_\theta$ that achieves a high return on the MDP given the initial state is one of the initial states of a demonstration $s_{i,0}$. It is critical to have the reverse curriculum as only applying a forward curriculum cannot work well in sparse reward settings due to exploration difficulty. If there do not

exist initial states $s_0 \in \mathcal{S}_{\text{init}}$ where it is easy to obtain any reward, the problem becomes very difficult and sample-inefficient, in addition to making it difficult to automatically generate a useful forward curriculum without relying on prior heuristics. Even when learning from demonstrations, if there are few demonstrations it still remains difficult to get the sparse reward due to the exploration problem and lack of state coverage in the few demonstrations, especially in complex tasks.

Thanks to a few demonstrations and reverse curriculum learning, the problem of 'no easy initial states' is alleviated, as we know that $\pi_\theta$ can at least perform well when starting from the initial states in the demonstrations. We only make a weak assumption that the initial state in the demonstrations we reverse-solved are close to states in $\mathcal{S}_{\text{init}}$, which is often the case in collected demonstrations. Then, using a forward curriculum, with RL we initially prioritize training on initial states where there is the most learning potential. These are often close to the initial states of demonstrations because the policy can still get some nonzero reward if the initial state is similar enough to one seen during reverse curriculum learning. Eventually, the forward curriculum will train the policy on a sufficient number of initial states in $\mathcal{S}_{\text{init}}$ and learn a policy that achieves high return on initial states sampled from $\mathcal{S}_{\text{init}}$, not just the demonstration initial states.

In order to maximize the outcome of each environment interaction, following PLR (Jiang et al., 2021) we prioritize resetting to initial states that are at the edge of the agent's abilities before resetting to states that are too difficult to solve from or states that are already easy to solve from. As PLR was originally designed for PPO, an on-policy algorithm, we make a simpler adaptation of PLR that works well and robustly for off-policy SAC and our environments.

First, we uniformly sample a set of $n$ initial states $\Lambda_{\text{train}} = \{s_{i,\text{init}}\}$ from $\mathcal{S}_{\text{init}}$. As policy $\pi$ after reverse curriculum learning has high return on initial states $s_{i,0}$ we add these to $\Lambda_{\text{train}}$. Our method uses a simpler score function to PLR that assigns three scores with higher scores leading to higher priority, and it does not have a separate seen vs unseen set of initial states. Define $q$ to be the fraction of episodes out of the last $k$ episodes that receive nonzero return starting from a sampled initial state $s_{i,\text{init}}$. If $q$ is 0, then assign a score of 2 to $s_{i,\text{init}}$. If $0 < q < \omega$ for a threshold $0 < \omega < 1$, assign a score of 3. If $q \geq \omega$, assign a score of 1. In order of decreasing priority, we sample initial states that sometimes receive return, then states that receive no return, then states that consistently receive return. Following PLR, we adopt a rank-based prioritization scheme which enables prioritization to be invariant to score scale, and we scale the importance of rank via a temperature value $\beta$, resulting in the score-prioritized distribution $P_S$. We also adopt the same staleness-aware prioritization $P_C$.

$$P_S(s_{i,\text{init}}|\Lambda_{\text{train}}, S) = \frac{\text{rank}(S_i)^{-1/\beta}}{\sum_j \text{rank}(S_j)^{-1/\beta}}, P_C(s_{i,\text{init}}|\Lambda_{\text{train}}, C, c) = \frac{c - C_i}{\sum_{C_j \in C} c - C_j} \qquad (1)$$

where $S_i$ are the scores assigned to initial state $s_{i,\text{init}}$, $c$ is the total number of episodes rolled out in training, and $C_i$ is the episode count at which initial state $s_{i,\text{init}}$ was last sampled. As $q$ is based on whether an episode received nonzero return or not, our score function is independent of scale of return of the environment under a optimal policy, making it a more generalizable scoring scheme. We follow the same intuition as PLR in that staleness prioritization ensures initial states that have not been sampled in a while get re-sampled in order to update their scores and ensure scores do not drift too far off their true value. The forward curriculum follows stages $\rho_0^f, \rho_1^f, ...$, advancing each time a new score is assigned. The linear combination of the score and staleness prioritzation distributions form the initial state distribution at each stage

$$P_{\rho_i^f}(s_{i,\text{init}}) = P_S(s_{i,\text{init}}|\Lambda_{\text{train}}, S) + P_C(s_{i,\text{init}}|\Lambda_{\text{train}}, C, c) \qquad (2)$$

For all experiments we use the same hyperparameters that construct the forward curriculum. We visually showcase how the forward curriculum progresses over the course of training in Fig. 2 via a toy example. While the initial policy starts off successful only on a narrow initial state distribution represented by the narrow green region, via the forward curriculum the policy prioritizes starting from initial states at the edge of what it is capable of and expands the green region until the majority of initial states are covered.

## 5 RESULTS

Through our experiments we aim to answer the following questions (1) How sample and demonstration efficient is RFCL compared to other model-free baselines? (2) What methods accelerate reverse curriculum learning? (3) How does the curriculums in RFCL enable success from significantly less demonstrations and robustness to the demonstration source?

For our experiments, we rigorously evaluate and compare our algorithm against several baselines across 3 robot environment suites for a total of 21 fully-observable environments with sparse rewards. The 3 environments suites are **MetaWorld** (Yu et al., 2019), **Adroit** (Rajeswaran et al., 2018), and **ManiSkill2** (Gu et al., 2023), see B.1 for details on each task. We use an absolute sparse reward where $+1$ is rewarded only on success states and 0 is given otherwise. The environments consist of a range of easy to challenging tasks and we benchmark our algorithm on the low demonstration regime, showcasing how RFCL is far more sample/demo efficient than prior approaches. We benchmark against 3 model-free baselines that also leverage demonstrations: (1) RLPD (Ball et al., 2023) is a well-tuned variation of SAC + demonstration data that at the time of writing achieves the state-of-the-art results on Adroit; (2) Jump Start RL (JSRL) (Uchendu et al., 2023) like RFCL leverages reverse curriculums, although they generate the curriculum with a offline trained guide policy. JSRL code is currently not available so we can only report on Adroit environments with numbers from their paper; (3) DAPG (Rajeswaran et al., 2018) uses a demo-augmented policy gradient for RL. (4) Cal-QL is a state-of-the-art offline-to-online RL method. For each training run, we select a random seed. The seed also determines which uniformly sampled demonstrations are used for training. All baselines use the same seeds and thus the same demonstration data.

### 5.1 MAIN BENCHMARK RESULTS

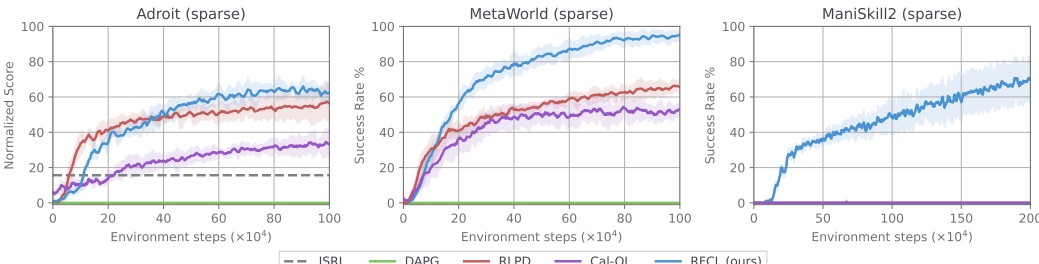

Figure 3: Mean success rate of algorithms for each environment suite across all tasks after 1M interaction steps with 5 demonstrations. Results are averaged within environment. Shaded areas represent 95% CIs over 5 seeds. The result show RFCL is significantly more performant and sample efficient compared to baselines. Note that RLPD and JSRL can normally achieve decent results but require many more demonstrations for harder tasks.

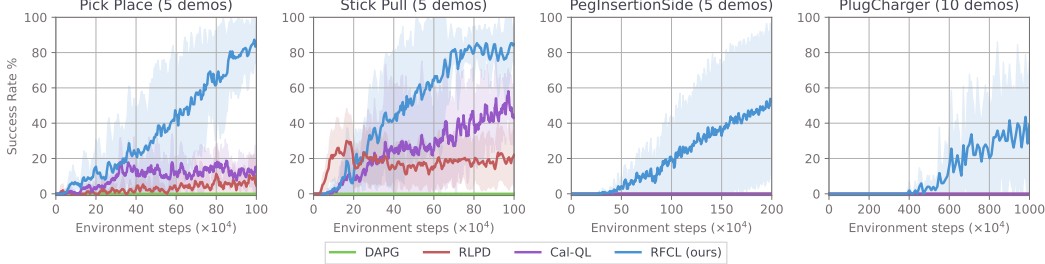

Figure 4: On the harder tasks, RFCL significantly outperforms baselines. PlugCharger from ManiSkill2 is not in the suite of tasks we test on usually as it is one of the most difficult environments, but RFCL can solve it given enough compute and demonstrations. Note PegInsertionSide and PlugCharger have high variance due to one seed failing for each, see Fig. 9 for the same graph with 30 seeds.

The main results are summarized in Fig. 3, showing how RFCL outperforms all baselines that leverage demonstrations across all benchmarks and tasks. There are significant improvements in ManiSkill2 and MetaWorld and minor improvements in Adroit. RFCL is the only method that is capable of achieving nonzero success on every environment within a reasonable compute budget. We argue that the critical reason why prior methods are unable to solve these tasks is because of the exploration bottleneck, which is more easily overcome via our method's state reset reverse curricu-

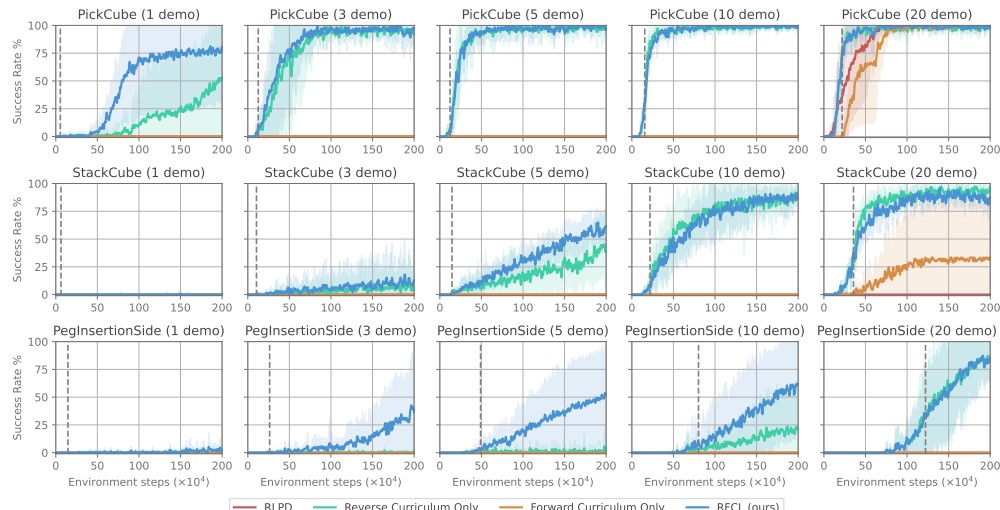

Figure 5: Mean success rate of algorithms on ManiSkill2 tasks after 2M interaction steps given varying amount of demonstrations to train on. Error bars represent 95% CIs over 5 seeds. Vertical gray lines indicate the average number of samples until the reverse curriculum completes. Reverse curriculum only is RFCL but instead of a forward curriculum in stage 2 we sample uniformly from the initial state distribution. Forward curriculum only skips stage 1 training entirely.

lum. Fig. 4 shows a selection of the most difficult tasks and demonstrate that RFCL is significantly better if the task is complex and difficult to explore. To visually see why these are more difficult see Appendix B.2. For a disambiguation of each environment's results, see Appendix A.1.

## 5.2 ABLATIONS

For all ablations, we benchmark on the ManiSkill2 environments as it is the most difficult set of environments due to high initial state randomization and precise manipulation requirements.

**# of Demonstrations:** RFCL provides a flexible approach to leveraging a wider range of demonstration dataset sizes and still being able to perform well as shown in Fig. 5. We observe that when there are very few demonstrations, the forward curriculum is important for improving performance. We see significant improvements on success rate for PickCube for 1 demonstration, for Stackcube for 5 demonstrations or less, and for PegInsertionSide for 10 demonstrations or less. The same conclusions can be drawn from the same demonstration ablation on Metaworld which we show in Appendix A.3. We can attribute this effect to the narrowness of the distribution of initial states the reverse curriculum trained policy performs well on. When given very few demonstrations covering a small set of initial states, the policy is successful on a narrow distribution of initial states. Without a forward curriculum, we are sampling initial states uniformly from $\mathcal{S}_{\text{init}}$, meaning a large majority of initial states are usually completely out of distribution and lead to 0 return and mostly useless exploration. With a forward curriculum, initial states that lead to nonzero return are prioritized, leading to sample-efficiency gains. Critically, we observe that RFCL still can solve difficult tasks that have high initial state randomization like PegInsertionSide from just a few demonstrations whereas the strongest baseline fails completely and can only solve PickCube when given $20\times$ the demonstrations RFCL needs to solve PickCube.

**Demonstration Source:** Adroit, Metaworld, and ManiSkill2 demonstrations are tele-operated, generated via scripts, and generated via motion planning respectively. For all these different kinds of demonstrations, RFCL can still solve the task showing that it is robust to demonstration source. This is exemplified by how human and motion planned demonstrations are often sub-optimal (with respect to a sparse reward maximization objective), exhibiting multiple modes in actions,

| Reverse Curriculum | Dynamic Timelimit | Steps ($\times 10^4$) |
|---|---|---|
| Uniform | ✓ | $> 100.0$ |
| Global | ✓ | $41.5 \pm 10.8$ |
| Per Demo | ✗ | $33.5 \pm 3.0$ |
| Per Demo | ✓ | $\mathbf{24.1 \pm 5.9}$ |

Table 1: Ablations on the reverse curriculum stage comparing number of environment steps until the agent can achieve $\sim100\%$ success rate from the initial states $\{s_{i,0}\}$ of 5 demonstrations in the ManiSkill2 suite. Results shown are 95% CIs over 5 seeds. Uniform choice failed to reach 100% within 1M interactions.

and possessing non-Markovian characteristics (Mandlekar et al., 2021). The robustness can be attributed to the fact that we are learning from a purely sparse reward objective via online interaction, and not perfectly imitating past demonstrations. The demonstrations primarily provide a means of overcoming exploration bottlenecks instead of leading to suboptima during training.

**Reverse Curriculum:** For stage 1 of training, alternative curriculums utilizing demonstration states for state resets are uniform and global. In a uniform "curriculum" we benchmark the uniform state reset method used in Nair et al. (2018) where they uniformly sample a random demonstration and state in the demonstration. In a global curriculum, instead of assigning each demonstration a start step value $t_i$, all demonstrations are assigned the start step value $T_i - u$ and we progress the curriculum of all demonstrations at the same time if the mean success rate of the past $v$ episodes exceeds a threshold. We perform a hyperparameter sweep over $v$ and reported the best results. We further ablate on the use of a dynamic timelimit. From the results in Table 1, we find that the uniform state reset approach when given just 5 demonstrations cannot get any reasonable results in 1M interactions. The per-demonstration reverse curriculum and dynamic timelimit both significantly improve the sample efficiency of the first training stage and learn a weak policy that can solve from $\{s_{i,0}\}$.

**Impact of Reverse and Forward Curriculums:** The toy experiment on a continuous state/action space pointmaze environment in Fig. 6 demonstrates the agent's performance throughout training. The agent can reset to any square not covered by the demonstration and the default initial state distribution is heavily biased to the squares far away from the goal, making exploration difficult. In order to succeed, a policy would need to explore around states along the demonstration as the state space is continuous. Notably, under the reverse forward cur-

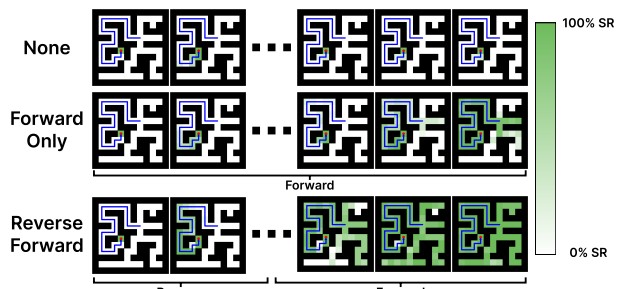

Figure 6: Heatmap of agent's success rate at each maze cell over the course of training, comparing three kinds of training: None (no curriculum / normal RL), forward curriculum only, and our method applying reverse and forward curriculums. Blue arrow is the demonstration provided. Red dot is the goal.

riculum the agent quickly learns to first perform well around states along the demonstration, after which it relies on the forward curriculum to quickly learn to solve from states all around the maze in a ever-growing frontier. Comparatively, without a curriculum under the same time budget the agent is unable to improve due to no initial state prioritization and difficult exploration. The forward curriculum works eventually thanks to prioritizing initial states closer to the demonstration and goal, but is less sample efficient compared to reverse forward due to exploration difficulty.

## 6 CONCLUSION

In this work, we introduce the RFCL algorithm that can solve complex tasks with significantly fewer demonstrations than before, including previously never-before solved sparse reward manipulation tasks. We show how the reverse curriculum trains an initial policy that solves the task from a narrow initial state distribution, and how the forward curriculum generalizes the policy to the full initial state distribution, enabling extreme demonstration and sample efficiency as shown in Fig. 3 and 5. We further motivate why the combination of a reverse and forward curriculum works, and introduce key methods that accelerate the reverse curriculum compared to alternatives.

A limitation of RFCL is the use of state reset and is thus generally restricted to training in simulation, relying on sim2real to be deployed in the real-world for robotics tasks. Despite this limitation, we argue sim2real is orthogonal research that can augment our research, and point to past work such as Chen et al. (2023) that succesfully perform sim2real on dexterous tasks and use state reset.

We hope our work provides a strong starting point for how far the capabilities of robot learning from demonstration methods can go if we fully leverage the advantages of simulation vs real world training. With the demo efficiency of RFCL, one can prioritize scaling up the diversity of tasks as opposed to the quantity of demonstrations. All code is open sourced on Github and are excited with how far the community can push when leveraging more properties of simulation.

## 7 REPRODUCIBILITY STATEMENT

All experiments on the RFCL method can be reproduced with given runnable scripts + docker images uploaded on https://github.com/stonet2000/rfcl

## 8 AUTHOR CONTRIBUTIONS

Stone Tao proposed the initial research ideas and experiments, in addition to writing the majority of code for training and evaluation. Arth and Tse-kai contributed to research discussions around various experiments and significantly to baselines. Hao Su provided advising and guidance to students on the research and proposed the toy experiment. All authors edited the manuscript.

## 9 ACKNOWLEDGEMENTS

Thank you to members of the Hao Su Lab for providing valuable feedback on the project. ST is supported in part by the NSF Graduate Research Fellowship Program grant under grant No. DGE-2038238.

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

# A  FULL ENVIRONMENT RESULTS

We show the training curves of our method compared to the primary baseline, RLPD, on all environments here without averaging across environments in Appendix A.1.

Because we are working with an extremely small number of demonstrations relative to the total number of demonstrations usually used with the tasks we tackle, there can be a high amount of variance in the result curves caused by the choice of demonstrations, not the algorithm itself. To disambiguate the source of variance, in Appendix A.2 we show results on ManiSkill2 on one of the harder environments PegInsertion where we have 2 sets of demonstrations and for each set we run 5 training runs. The 2 sets of demonstrations are the same ones sampled for the main results shown in Figure 3.

Finally we show some additional experiments with a demo ablation on MetaWorld in Appendix A.3 and compare against a baseline that uses state reset by Nair et al. (2018) on one of their tasks in Appendix A.4 (their open sourced code does not use state reset so we can only test on their environment instead of benchmarking their algorithm on the more common robotics benchmarks like Metaworld).

## A.1  RESULTS WITH RANDOM DEMONSTRATIONS

All algorithms are given 5 randomly sampled demonstrations. We further mark the average step at which the reverse curriculum stage completes training with a vertical, dashed gray line. The bolded lines represent the mean values, and the shaded areas represent 95% CIs over 5 seeds. Note that each seed uses the same demonstrations. Figures 7 and 8 show results on each environment in MetaWorld and Adroit respectively. The demo ablation curve in Fig. 5 in the main paper shows results for ManiSkill2. We further explored just how variable demonstations are by running 30 seeds on the PegInsertion task with each seed using different sets of 5 demonstrations in Fig. 9. The figures show that we are the only method to succeed on all environments, significantly beating baselines. Furthermore, empirically our method is generally quite robust to choice of demonstrations as shown in in Fig. 9. Only 1 seed out of 30 failed to get above 80% success rate after 6M interactions. Note that the robustness may vary between environments, and our results in the main figures do include the seeds where nearly no success was obtained that occured. We believe there could be interesting future work in investigating what kind of demonstrations are easier to learn from using RFCL.

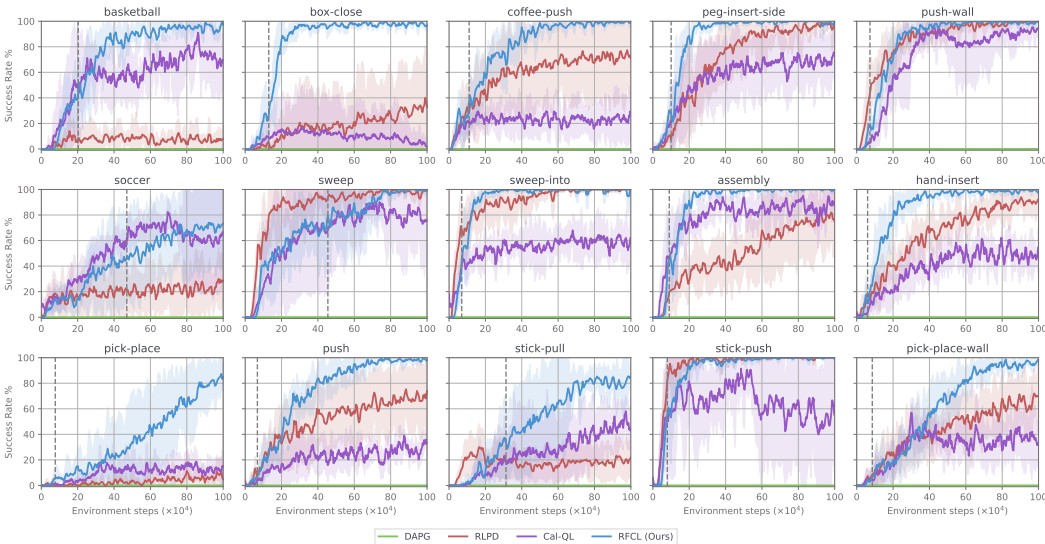

Figure 7: Metaworld Results with random demonstrations. Results averaged over 5 seeds with shaded area representing 95% CIs. Every single seed succesfully finished the reverse curriculum and the success rate trended upwards.

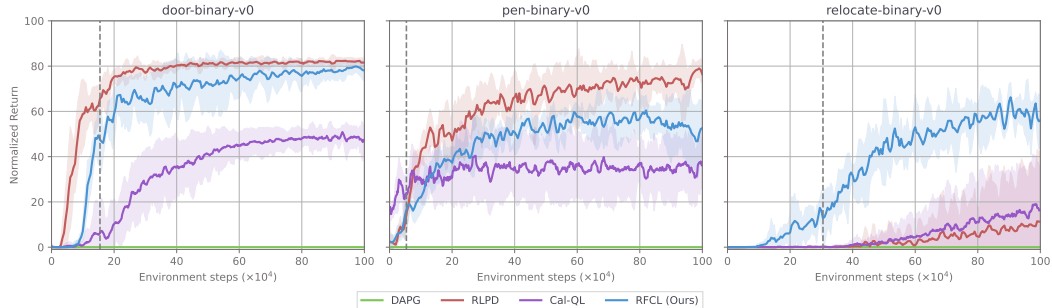

Figure 8: Adroit Results with random demonstrations. Results averaged over 5 seeds with shaded area representing 95% CIs. Every single seed succesfully finished the reverse curriculum and the success rate (not shown) reached near 100%. The normalized return of RFCL plateaus just below the state-of-the-art method RLPD for the two easier environments, but significantly outperforms RLPD and Cal-QL on the harder Adroit Relocate environment.

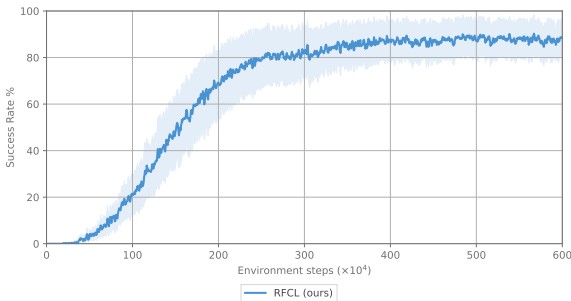

Figure 9: ManiSkill2 PegInsertion results with random demonstrations. Results averaged over 30 seeds with shaded area representing 95% CIs. The results appear much better than the main figures due to the noise caused by hard-to-learn-from demonstrations being mostly cancelled out. Here, only 1 out of the 30 seeds failed (the same seed from the main results that failed.

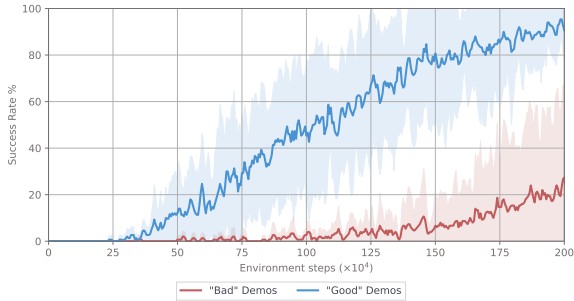

Figure 10: ManiSkill2 PegInsertion results with fixed demonstration sets. "Bad" Demos correspond to runs using the only set of demonstrations that led to no success in the main results in Fig. 4. "Good" Demos is randomly chosen to be one of the other set of demonstrations used in the main results. Results averaged over 5 seeds with shaded area representing 95% CIs.

## A.2 RESULTS WITH FIXED DEMONSTRATIONS

We pick 2 of the 5 sets of demonstrations used for training shown in Figures 3 and 4, one of which was succesful and solved PegInsertionSide rather fast, and the other demonstration set being the one where there was no success after training for 2M interactions. We then run 5 seeds on both sets of demonstrations and show the results in Fig. 10. We observe that the failed set of demonstrations is indeed harder to learn from, although not impossible as 2/5 seeds can solve the task using that set of

demonstrations while 3/5 seeds still has around 0 success after 2M interactions. For the successful set of demonstrations, all seeds solved the task relatively quickly.

## A.3 METAWORLD DEMO ABLATION

Fig. 11 is another set of demo ablations on the Metaworld suite comparing RFCL with reverse curriculum only and shows that when there are fewer demonstrations, the forward curriculum enables more sample efficiency. The conclusions drawn from the main demo ablation on ManiSkill2 in Fig. 5 can also be drawn here.

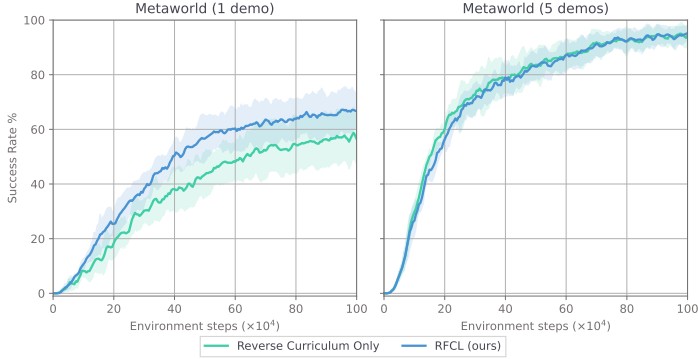

Figure 11: Metaworld Results with varying number of demonstrations and comparing RFCL vs Reverse curriculum only. Results averaged over 5 seeds with shaded area representing 95% CIs.

## A.4 COMPARISON AGAINST PREVIOUS STATE RESET METHODS

Fig. 12 compares our RFCL method with the state reset method proposed by Nair et al. (2018) on the difficult Stack 3 task, which tasks the agent to stack 3 blocks into a tower. We label their method as Nair et. al (2018). Our approach vastly outperforms their method by being able to solve the task in less than 10 million samples with just 20 demonstrations whereas Nair et al. (2018) requires more than 100 million samples using 100 demonstrations to just get a 39% success rate and has 0 success even after 50 million samples (see their paper for their exact success rate curves). We note that while RFCL performs better, this comparison is a little unfair as the core RL algorithms used are different between papers (hence why this figure is in the appendix). For a more fair comparison we benchmarked the state reset strategy Nair et al. (2018) uses in Table 1 while keeping the rest of RFCL the same, which still shows that our per-demo reverse curriculum is far more sample efficient compared to alternatives

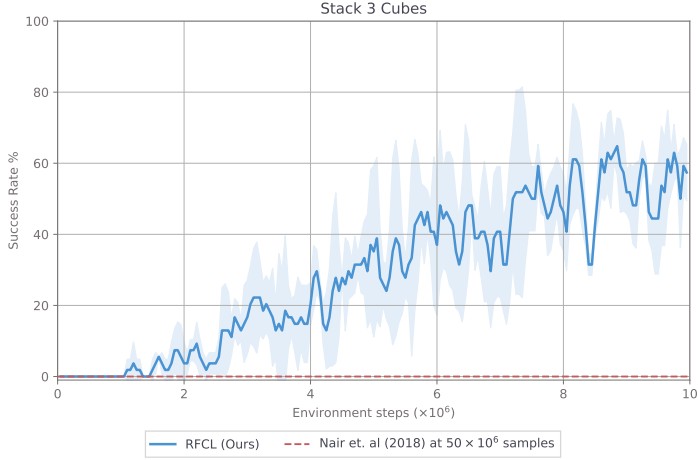

Figure 12: Stack 3 results of RFCL when given 20 random demonstrations. Results averaged over 5 seeds with shaded area representing 95% CIs.

## B  TASK DETAILS AND VISUALIZATIONS

### B.1  TASK DETAILS

Below is the observation dimensions, action dimensions, and task horizons of all benchmarked tasks. For ManiSkill2 and Metaworld, we use the more robust and difficult metric of measuring success at the end of episodes as opposed to allowing success and then failure (e.g. stopping an episode upon success being reached). Following recent past work on Adroit environments, we measure a normalized score which is equal to the percent of timesteps in an episode where the agent is in a success state. For all tasks we do not use any action repeat.

| Task | Observation Dimensions | Action Dimensions | Horizon |
| --- | --- | --- | --- |
| Adroit Door | 39 | 28 | 200 |
| Adroit Pen | 45 | 24 | 100 |
| Adroit Relocate | 39 | 30 | 200 |

Table 2: Adroit Tasks: We consider the same 3 Adroit tasks and task horizons as our baselines JSRL and RLPD. The controller used to control the Adroit hand is the default one in the benchmark, which controls each finger joint and the hand position with delta actions.

| Task | Observation Dimensions | Action Dimensions | Horizon |
| --- | --- | --- | --- |
| All Tasks | 39 | 4 | 200 |

Table 3: Metaworld Tasks: We consider the most difficult 15 Metaworld tasks following the categorization by Seo et al. (2022). The controller used to control the robot end-effector is the default one in the benchmark, which controls the 3D position and gripper position of the end-effector with delta actions.

### B.2  TASK INITIAL START STATES AND CATEGORIZING DIFFICULTY

In Figures 13 to 18 we show visually what randomly sampled start states from the true start state distribution $\mathcal{S}_{\text{init}}$ look like. We further note which of the environments have large amounts of randomization, which environments do not, and where the randomization comes from, in addition to briefly detailing where the complexity of the task comes from that makes it difficult to solve. Our conclusion with regards to which environments are difficult in MetaWorld is corroborated by the categories used by Hansen et al. (2023); Seo et al. (2022).

Notably we observe that environments that are complex and have a large amount of initial state randomization tend to need more compute or demonstrations to solve while those with less randomization can get by with much less demonstrations. For example, Adroit Door has little to no

| Task | Observation Dimensions | Action Dimensions | Horizon |
|---|---|---|---|
| PickCube | 51 | 7 | 100 |
| StackCube | 55 | 7 | 100 |
| PegInsertionSide | 50 | 7 | 100 |
| PlugCharger | 53 | 7 | 100 |

Table 4: Maniskill2 Tasks: We consider 4 tasks of various difficulty, with PickCube being the easiest (and solvable by baselines when given enough demonstrations) and PegInsertionSide/PlugCharger being the most difficult. PegInsertionSide and PlugCharger have never been solved before under sparse rewards and few demonstrations. The controller used to control the robot end-effector is the called pd-ee-delta-pose, one of the defaults in the benchmark, which controls the pose and gripper position of the end-effector with delta actions.

randomization and is easily solved by RFCL and the RLPD baseline with just 1 demonstration. On the other end, ManiSkill2 PegInsertionSide and PlugCharger are complex and have a lot of randomization and can only be reliably solved by our method RFCL when given enough demonstrations and/or more compute. The randomization magnitude can stem from higher dimensions (e.g. more objects have randomized shape and positions) and/or stem from a wider range of random sampling.

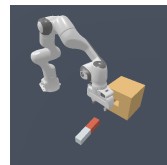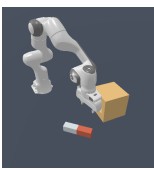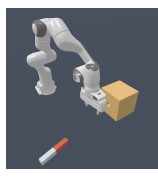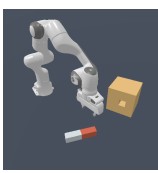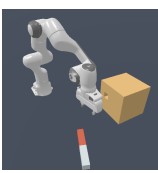

Figure 13: ManiSkill2: PegInsertionSide. High randomization: peg length, peg width, peg pose, block pose, hole in block position, robot position. Complex: Small clearance between peg and hole, requires learning to grasp and insert precisely of different shapes from highly randomized poses

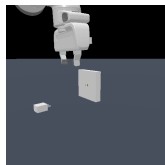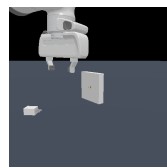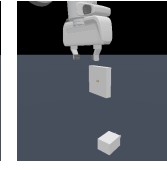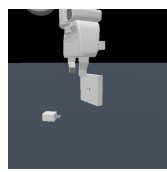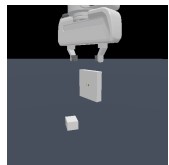

Figure 14: ManiSkill2: PlugCharger. High randomization: charger pose, receptacle pose, robot position. Complex: Very small clearance (much smaller than any other environment benchmarked in this paper) between the charger head and receptacle space, requires learning to grasp and insert very precisely

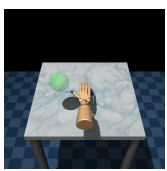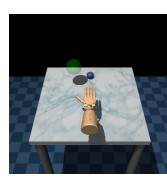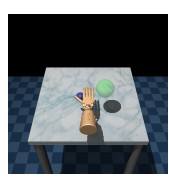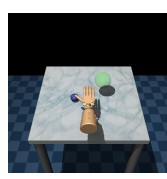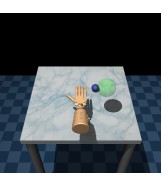

Figure 15: Adroit: AdroitHandRelocate. High randomization: ball position, goal position. Complex: Difficult dynamics with regards to grasping a ball with a high-dimensional robot

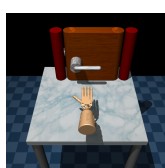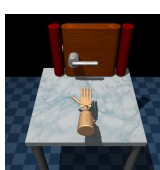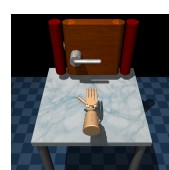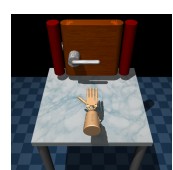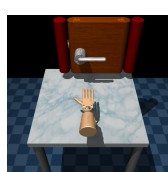

Figure 16: Adroit: AdroitHandDoor. Low randomization: door position. Simple: While robot hand is high-dimensional, the door barely changes position

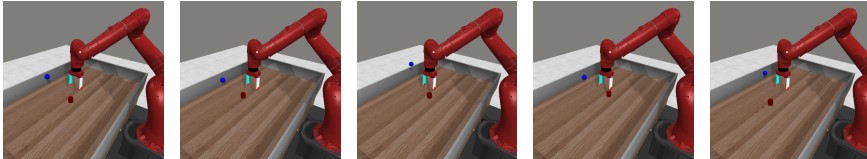

Figure 17: MetaWorld: Pick Place. High randomization: block position, goal position. Complex: Highly randomized positions of goal and object, and object is small leaving little room for error.

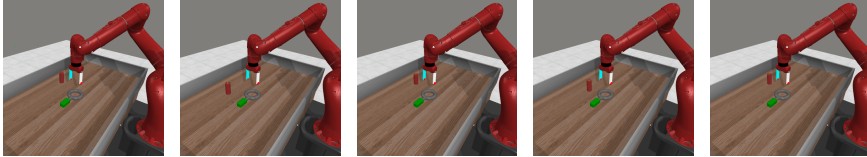

Figure 18: MetaWorld: Assembly. Low randomization: cylinder position. Simple: Only the cylinder position is randomized and to only a small range, all other objects are initialized in the same place.

### B.3 TASK REVERSE CURRICULUM

Below are samples of various states of the automatically generated per-demonstration reverse curriculum over the course of training for some of the more difficult tasks. Note that during the reverse curriculum stage of training, the reverse step size is small and the distance between adjacent stages of the curriculum are smaller than shown here.

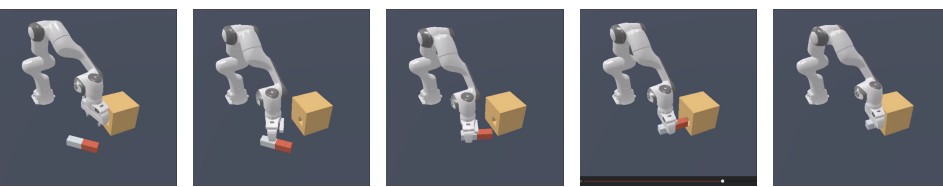

Figure 19: ManiSkill2: PegInsertionSide

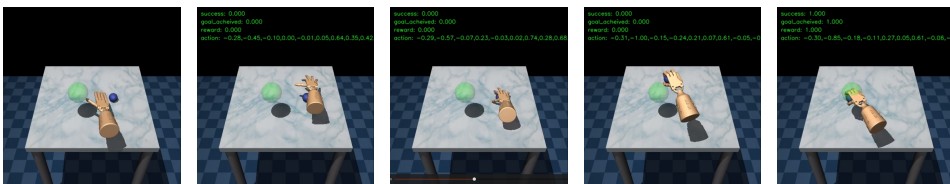

Figure 20: Adroit: Relocate

### B.4 TASK SOLVES

For videos of example solves of these environments produced by our method RFCL, see our website at https://reverseforward-cl.github.io/

### B.5 BASELINE SELECTION METHODOLOGY

As we are proposing a learning-from-demonstrations model-free RL algorithm that is more efficient in terms of samples and demonstrations, we select state-of-the-art model-free RL baselines that can use demo data with code.

We select RLPD (Ball et al., 2023) as its the current state of the art method on the Adroit suite of tasks for learning from demos. It is the best baseline we have as well. As the RLPD paper shows it out-performs standard SAC + adding demo data to the online buffer, we do not include a SAC baseline.

We also select Jump Start RL (Uchendu et al., 2023). Although it does not achieve state of the art results on Adroit, the concept of a reverse curriculum shares similarities with our proposed RFCL method.

We further select DAPG (Rajeswaran et al., 2018) and Cal-QL (Nakamoto et al., 2023) as they once achieved state-of-the art results on some tasks for learning from demos algorithms.

There are many on-policy algorithms including the popular PPO (Schulman et al., 2017) baseline that could be benchmarked. However, it is well known that off-policy algorithms like SAC are much more sample efficient than on-policy algorithms, and it would be unfair to compare against on-policy methods as they would do poorly. The recent Sequential Dexterity (Chen et al., 2023) method in theory would be good to compare against as they are one of the few papers with code that use state resets, however they use PPO and rely on human engineering to split a task into subtasks which is infeasible to do on the 20+ tasks we test on.

There are also model-based methods like MoDem (Hansen et al., 2023) that are very sample efficient thanks to the off-policy setup and model-based approach. We do not compare against MoDem directly as they use world models, but we believe it would be interesting future work to investigate reverse forward curriculums with world models for even better sample efficiency.

Finally, while we cite a number of past methods that use state resets, we currently are unable to benchmark any of them as the majority do not have code and/or have results on tasks that are not open-sourced. We qualitatively comment on key differences in the related works section Sec. 2 and benchmark the uniform state reset strategy employed by Nair et al. (2018) and show that strategy performs worse compared to our per-demo reverse curriculum state reset approach in 1. We further show results of RFCL on a task that Nair et al. (2018) tests on in Appendix A.4.

## C  HYPERPARAMETERS

| Hyperparameter | Value |
|---|---|
| **RL Hyperparameters (Soft-Actor-Critic)** | |
| Discount factor ($\gamma$) | 0.99 (Adroit and MetaWorld tasks) |
| | 0.99 (MS2 PegInsertionSide and Plugcharger) |
| | 0.9 (All other MS2 tasks) |
| Update to Data Ratio | 10 |
| Actor Update Frequency | 20 |
| Seed Steps | 5,000 |
| Replay Buffer Capacity | 1,000,000 |
| Batch Size | 256 |
| Number of Critics | 10 |
| Number of Sampled Critics | 2 |
| Polyak Averaging Coefficient ($\tau$) | 0.005 |
| Initial Temperature | 1.0 |
| Learnable Temperature | True |
| Total Interactions / Samples | 1,000,000 |
| Offline buffer to online buffer sample ratio | 50:50 |
| **Networks and Optimization** | |
| Network Shape of Actor and Critic (MLP) | [256, 256, 256] |
| Layer Norm | Applied after every layer of critic only |
| Activation | Rectified Linear Unit (ReLU) |
| Actor Learning Rate | 3e-4 |
| Critic Learning Rate | 3e-4 |
| Actor and Critic Optimizers | Adam |
| **Environment and Data** | |
| Number of demonstrations | 5/25 human teleoperated |
| | demonstrations for Adroit |
| out of total demonstrations available | [1-20]/1000 motion planned |
| | demonstrations for ManiSkill2 |
| | 5/$\infty$ scripted demonstrations for MetaWorld |
| Reward Function | Sparse (+1 on success, 0 otherwise) |
| Action Repeat | 1 |
| Episode Horizon | 100 (ManiSkill2 and Adroit Pen) |
| | 200 (Adroit and MetaWorld Tasks) |
| Observation Type | State |
| **Reverse Curriculum** | |
| Reverse Step Size ($\delta$) | 8 for Adroit and MetaWorld |
| | 4 for ManiSkill 2 |
| Per-trajectory Curriculum Buffer Size ($m$) | 3 |
| Demonstration Horizon to Episode Horizon Ratio ($\phi$) | 3 |
| **Forward Curriculum** | |
| Forward Curriculum Score Threshold ($\omega$) | 0.75 |
| Forward Curriculum Buffer Size ($k$) | 5 |
| Forward Curriculum Score Temperature ($\beta$) | 0.1 |
| Forward Curriculum Ranking Strategy | Rank |
| Forward Curriculum Staleness Coefficient | 0.1 |
| Forward Curriculum Number of Seeds/Levels ($n$) | 1000 |

Table 5: **RFCL** sample-efficient variation of hyperparameters. These are the ones used to generate all figures and results. Highlighted in blue indicates hyperparameters introduced by this paper, which are for the automatic construction of reverse and forward curriculums.

The non-highlighted hyperparameters are standard ones used in Soft-Actor-Critic with a Q-ensemble or Prioritized Level Replay (PLR). Furthermore, while a discount of 0.99 could be used for all tasks, we opt to use 0.9 for some ManiSkill2 tasks as it seems neither our method nor any of the baselines can solve the tasks with a discount of 0.99 easily in 1M interactions when given significant amounts of demonstration data. Moreover, while it is strange only 1 Adroit task (Pen) has a smaller horizon of 100, recent past work (Uchendu et al., 2023; Ball et al., 2023) use this specific horizon so we copy them for fair comparison.

Finally, for just the PlugCharger environment, we make a few modifications to make it work relatively more efficiently. We use a discount of 0.95, batch size of 512 to improve sample efficiency a little. Importantly, we rescale the action space to be 125% the magnitude of the actions in the set of demonstrations used during training as the demonstration actions of PlugCharger on average are very small in magnitude, suggesting the full action space of the environment is unrealistic and pose great difficulty to exploration. This is a fairly reasonable method to make the problem easier without relying on heavy heuristics, just introducing some bias about the magnitude of actions. We observe this rescaling not only accelerates learning but produces more reasonable policies that take smaller actions and are likely easier to transfer to real. We only use the action rescaling for PlugCharger tests, leaving all other environments as they were, although we believe this would likely accelerate learning for other environments requiring high precision like PegInsertionSide.

## D  WALL TIME EFFICIENT RESULTS

We note that while it is standard in the field of robot learning and reinforcement learning to compare performance against the number of environment samples, for practical purposes it is also important to have fast wall-time performance. This is made more critical if one's direction of research is towards sim2real of which then environment sample time is generally fast. With fast environment sample times, it is more wall-time optimal to increase the number of environment samples and lower the relative number of gradient updates, which is done with our wall time efficient set of hyperparameters. The only changes are to use a much smaller update to data ratio of 0.5 and increase actor update frequency to 1, essentially for e.g. every 32 environment steps we update the critic and actor 16 times each. Moreover, the fast hyperparameters run 8 parallel environments with each taking 4 steps at a time, after which then 16 gradient updates are performed. Training can go even faster if 32 parallel environments are used.

We did not heavily tune these hyperparameters since they were already quite fast, but we note that depending on environment speed, it may be optimal to tune the update to data ratio accordingly.

Time in Table 6 is reported in GPU minutes representing the time until the running average success rate of the last 5 evaluations reaches 90%. Experiments all ran on a RTX 2080 GPU. This is a very loose evaluation as it includes the non-trivial amount of time spent running test evaluation in addition to actual training, but it's clear that simply tuning a few hyperparameters can easily improve training time.

| Task | Wall Time Efficient Hyperparams | Sample Efficient Hyperparams |
|---|---|---|
| Metaworld Soccer | 47min | 143min |
| Metaworld Stick Pull | 38min | 104min |
| Metaworld Pick Place | 39min | 180min |

Table 6: Training time of select environments using different hyperparameters.

