# OpenReview forum: "Reverse Forward Curriculum Learning for Extreme Sample and Demo Efficiency"
_ICLR.cc/2024/Conference — ICLR 2024 poster_

### Official Review · Reviewer_gfCC · 2023-11-01

**Soundness:** 3 good
**Presentation:** 3 good
**Contribution:** 3 good
**Rating:** 6
**Confidence:** 3

**Summary:**

In this study, a novel methodology is proposed, combining reverse and forward curricula. The authors suggest resetting from demonstrations to effectively perform exploration when access to a limited number of demonstrations is available. The reverse curriculum starts from positions backward from the goal point using demonstrations, while the forward curriculum is executed through score-based prioritizing, allowing the starting point to have an appropriate level of difficulty. As a result, it shows improved performance compared to baseline methods that leverage demonstrations.

**Strengths:**

- The proposed method, combining reverse-forward curriculum approach is novel.

- The learning curves presented in figures 3 and 4 demonstrate improved performance compared to the baselines, and the figure in 5 suggests that using both the reverse and forward curriculum with a limited number of demonstrations is beneficial.

**Weaknesses:**

- The assumption of resetting from demonstrations is a strong one and only applicable in simulation environments. While this study proposes an efficient way to leverage one or more demonstrations through the reverse curriculum, I still consider it to be a strong assumption.

- The study may appear to be not significantly different from consecutively performing Jump-Start RL and PLR.

- While it compares to various baselines, many of them seem to be algorithms that do not assume state resets or use curricula.

**Questions:**

- In Figure 6, it seems that 'reverse' and 'forward' are only applicable to 'reverse-forward,' and not to 'none' and 'forward only,' which might be confusing.
- Which algorithms among the compared baselines require the State Reset assumption?
- If possible, could you please explain the reason for the initial high performance of RLPD in the stick pull task in Figure 4, followed by a drop?

---

> ### Author Response · Authors · 2023-11-12
> **Response to Reviewer gfCC [1/2]**
>
> Thank you for the review and raising the concerns, we are happy to see you believe our approach is novel. We address concerns below:
>
> > The assumption of resetting from demonstrations is a strong one and only applicable in simulation environments. While this study proposes an efficient way to leverage one or more demonstrations through the reverse curriculum, I still consider it to be a strong assumption.
>
> We agree that this is a strong assumption, as is the case for many other papers in our research area focusing on robot learning ([1], [2]). We argue however that work on sim2real is orthogonal to our research and thus investigating the most efficient algorithms in simulation is of great value. Papers like [1] have shown using sim2real can still achieve some fairly dexterous capabilities on real robots despite training only in simulation. Additionally, another use case that arises from using simulation is having an algorithm that trains much faster and uses less demos in simulation can help generate more diverse simulation data to be used with limited real world data to help scale up training data for large-scale robotics models.
>
> > The study may appear to be not significantly different from consecutively performing Jump-Start RL and PLR.
>
> In general our reverse curriculum is fairly different to Jump-Start RL (JSRL). JSRL suffers from exploration problems in sparse reward settings. We alleviate some significant exploration problems by efficiently leveraging state resets. Moreover, we construct a unique per-demo curriculum and introduce a few techniques like dynamic time limits that JSRL does not have which enable faster training as shown in ablations.
>
> > While it compares to various baselines, many of them seem to be algorithms that do not assume state resets or use curricula… Which algorithms among the compared baselines require the State Reset assumption?
>
> JSRL is the only baseline that uses some form of reverse curriculum with comparable results (they test on proper benchmarks that have code). We do cite several past methods that use reverse curricula and state reset but they do not have code and often test on simple tasks (e.g. maze navigation) outside of robotics that also have no code. As an alternative, we perform an ablation testing the use of a uniform state reset that methods like [2] use in section 5.2 (Table 1) instead of the reverse curriculum RFCL uses. Across ManiSkill2 tasks, uniform state reset is incapable of reaching high success rates within 1M environment interactions when evaluating on initial states $s\_{i, 0}$ in the 5 demonstrations given. Finally, methods like [3] use curriculums but assume access to human-engineered curriculums which is a stronger assumption so they are hard to compare fairly.
>
> We have only found one open-sourced baseline that claims to use state reset which is the one by Nair et. al. [2]; however we could not get their code working correctly as upon further inspection, their code does not use state resets. We will run our RFCL method on the environments Nair et. al. [2] test on and compare against their written results obtained via state resets and report here. [1] also uses state reset and has some code but is not optimizing to be sample efficient as it uses PPO so it would be unfair to compare against (in addition to relying on a number of human designed heuristics).
>
> For now, to compare against [2], you can look at the PickCube results for RFCL in Fig. 5 which is a very similar environment to the Pick and Place task [2] uses. The main difference is our task has a larger action space as we can control the pose of the end-effector (necessary to help grasp rotated cubes) whereas in [2] you are fixed to controlling just the position of the end-effector to grasp cubes with fixed rotation. RFCL with just 5 demos is capable of achieving a high success rate after 250k samples whereas [2] needs about 500k+ samples despite using 100 demonstrations and solving a easier task.
>
> [1] Chen et. al, “Sequential Dexterity” CoRL, 2023
>
> [2] Nair et. al, “Overcoming exploration in reinforcement learning with demonstrations” ICRA, 2018
>
> [3] Li et. al, “Towards practical multi-object manipulation using relational reinforcement learning” ICRA, 2020

---

> ### Author Response · Authors · 2023-11-12
> **Response to Reviewer gfCC [2/2]**
>
> > In Figure 6, it seems that 'reverse' and 'forward' are only applicable to 'reverse-forward,' and not to 'none' and 'forward only,' which might be confusing.
>
> This is actually the correct interpretation. We are simply marking when the reverse and forward curriculum are active. We have added an extra bracket under the second row indicating that the forward curriculum is constantly running for that ablation.
>
> > If possible, could you please explain the reason for the initial high performance of RLPD in the stick pull task in Figure 4, followed by a drop?
>
> Unfortunately it is difficult to say why this has occurred, although unlikely, maybe some data corrupted when uploading training results. We are currently rerunning RLPD with the same 5 seeds plus 5 more seeds on the stick pull task and will re-record results, it might also just be some strange noise. Overall however the conclusions remain the same about RFCL's performance relative to RLPD.
>
> We hope our response (plus forthcoming new results) addresses all your concerns. In light of these clarifications, would you be able to raise your score? We would be happy to address any more issues you have.

---

> > ### Comment · Reviewer_gfCC · 2023-11-20
> >
> > Regarding "JSRL suffers from exploration problems in sparse reward settings. We alleviate some significant exploration problems by efficiently leveraging state resets." could you provide specific details on which differences lead to variations in exploration performance in sparse reward settings?
> >
> > In other words, I understand that JSRL leverages the implicit distribution naturally occurring through deploying a guiding policy without explicitly controlling initial state. With the proposed method introducing the assumption of state reset, can it provide additional insights beyond the performance gains that arise from this assumption (state reset)?
> >
> > All other questions have been addressed.

---

> > > ### Author Response · Authors · 2023-11-20
> > > **Response by Authors**
> > >
> > > Thanks for the additional comments, we are glad we have been able to address the other questions
> > >
> > > > Regarding "JSRL suffers from exploration problems in sparse reward settings. We alleviate some significant exploration problems by efficiently leveraging state resets." could you provide specific details on which differences lead to variations in exploration performance in sparse reward settings?
> > >
> > > JSRL relies in the guide policy to perform the initial rollout and you are correct that this creates a distribution of initial states that the exploration policy in JSRL can start from. The problem here is that the guide policy is trained with offline RL and then frozen when the exploration policy starts to be trained. Offline RL has difficulty getting accurate value functions when given limited data. Even if you add some online data to help train the guide policy, because of sparse reward most of that online data is less informative as they often have reward label of 0. With fewer demonstrations offline RL performs even worse, and this is visible in the baselines JSRL tests on such as IQL. Given that when there are few demonstrations the offline RL trained guide policy performs poorly, it is very unlikely the guide policy can rollout to reach initial states near the goal states, meaning the exploration policy will still have to deal with all the problems of sparse reward as it initializes far from the goal states.
> > >
> > > The use of state resets allows the agent to directly be initialized near the goal states and thus have a high likelihood of obtaining reward and useful data during online training. Offline RL trained guide policy given few demonstrations will almost never be able to rollout to states close to the goal.
> > >
> > > A practical example can be seen with Adroit Relocate. JSRL given 5 or fewer demos gets a reported 0 success, meaning the guide policy likely has never rolled out to states where the robot hand has grasped the ball and reached near the goal. The exploration policy then still has the same sparse reward problem as all the online data collection being performed never yields any nonzero return due to the difficult exploration of a high dimensional robotics task. In contrast, our state reset directly resets the agent to the state in the demo where the robot hand is grasping the ball and the ball is at the goal, so this agent in RFCL can collect data with reward learn more efficiently.
> > >
> > > > ...  With the proposed method introducing the assumption of state reset, can it provide additional insights beyond the performance gains that arise from this assumption (state reset)?...
> > >
> > > Sorry, we did not quite understand what this statement means but we hope the earlier response here may answer it.

---

> > > > ### Comment · Reviewer_gfCC · 2023-11-21
> > > >
> > > > Thank you for the response that addresses my questions. I would like to raise the score.

---

> > > > > ### Author Response · Authors · 2023-11-21
> > > > >
> > > > > Thank you for raising your score! We really appreciate the early engagement with us to clarify details.

---

### Official Review · Reviewer_Rzng · 2023-11-01

**Soundness:** 3 good
**Presentation:** 2 fair
**Contribution:** 3 good
**Rating:** 8
**Confidence:** 4

**Summary:**

In this paper the authors show the benefits of first using a reverse curriculum on demonstrations (of simulated RL tasks with sparse rewards) followed by a forward curriculum that expands the set of initial states (from which the agent can achieve the task). In many different simulated RL tasks, it is shown that the method can achieve quite significant boosts in sample efficiency, and can sometimes succeed in tasks which other methods completely fail. Ablation studies show that the method is robust to the number of demonstrations and that the reverse-forward curriculum seems to be the best among various other choices.

**Strengths:**

* The paper convincingly shows that their method and their particular choice of curriculum leads to significantly better performance in many different RL tasks with sparse reward structure.

* Ablation studies are well done and cover significantly the possible variations.

* Figures captions and plots are well done, as well as the visualizations in the website.

**Weaknesses:**

* The paper could benefit from an algorithmic summary. Algorithmic decisions not summarized succinctly by equations or in algorithmic form, but by verbose descriptions make the method look more 'alchemical' than it need be. e.q.

"Define q to be the fraction
of episodes out of the last k episodes that receive nonzero return starting from a sampled initial state
si,init. If q is 0, then assign a score of 2 to si,init. If 0 < q < ω for a threshold 0 < ω < 1, assign
a score of 3. If q ≥ ω, assign a score of 1."

The actual numbers chosen detract from the concept of rejecting samples based on e.g. the expected return of exploration.

**Questions:**

* Please do not use the word 'extreme', you used it several places throughout the paper.

* "In practice, the observations used by the policy πθ may be different from the actual environment state but for simplicity in this
paper state also refers to observation."
> So you don't consider noisy feedback or POMDPs? This should be mentioned clearly in the introduction. As the method shows significant improvement in learning curves, it is vital to indicate when/where we expect them to hold and where they would fail.

* "In both stages, we use the off-policy algorithm Soft Actor Critic (Haarnoja et al., 2018) with a Q-ensemble (Chen et al., 2021b), "
> What happens if you use another RL algorithm? Does it make a big difference? Which other methods, competitive to SAC, could you use?

* Minor comment: " As a result, a curriculum for each demonstration is necessary as opposed
to a curriculum constructed from all demonstrations as done in prior work in order to ensure noisy
information arising from the multi-modality of demonstrations do not impact the reverse curriculum
of each demonstration as much." -> Too long sentence.

* You use \phi before you introduce it, and I didn't get what it's supposed to mean?

* "In this manner, initial states that sometimes receive
return are prioritized the most, then initial states that receive no return, then initial states that are
consistently receiving return." -> Not a very clear sentence, do you want to use 'than' instead?

* Would be nice to discuss why the methods compared against were chosen out of all the possible RL algorithms out there (maybe in an appendix?) Would the others be unsuitable for the task (e.g. on-policy, not suitable for demonstrations etc.)

==== POST-REBUTTAL ====
* My score remains the same, I think this paper should be accepted, as it has good results and detailed ablations.
* Assumptions/limitations of the method (e.g. full observability, restricted to simulations, requires demonstrations) are mentioned throughout the paper and is not a deal-breaker. As mentioned in the rebuttal, just improving simulation efficiency is also a contribution.

---

> ### Author Response · Authors · 2023-11-13
> **Response to Reviewer Rzng**
>
> Thank you for the high rating, we are happy to see that you found the results and figures convincing. We address your concerns below:
>
> > The paper could benefit from an algorithmic summary. Algorithmic decisions not summarized succinctly by equations or in algorithmic form, but by verbose descriptions make the method look more 'alchemical' than it need be.
>
> Thank you for bringing this up, we will look to revise this to be better as we work through the rebuttals.
>
> > Please do not use the word 'extreme', you used it several places throughout the paper.
>
> Revised!
>
> > "In practice, the observations used by the policy πθ may be different from the actual environment state but for simplicity in this paper state also refers to observation."
> So you don't consider noisy feedback or POMDPs? This should be mentioned clearly in the introduction. As the method shows significant improvement in learning curves, it is vital to indicate when/where we expect them to hold and where they would fail.
>
> In our revision we have clarified in the introduction that we only benchmark on fully observable environments (as also done by benchmarked baselines).
>
> > "In both stages, we use the off-policy algorithm Soft Actor Critic (Haarnoja et al., 2018) with a Q-ensemble (Chen et al., 2021b), "
> What happens if you use another RL algorithm? Does it make a big difference? Which other methods, competitive to SAC, could you use?
>
> This is a great question. This could be an interesting line of future work to investigate if RFCL is actually RL algorithm agnostic. Our initial motivation to use an off-policy algorithm as they are known to be more sample-efficient than on-policy algorithms in robotics benchmarks and we compare against baselines over sample-efficiency. We use SAC (with Q-ensemble) in particular as empirically it has the best performance in the robotics benchmarks we test on. Algorithms like PPO can also be used, especially if the interest is in training wall-time and not sample-efficiency. However one may need to rethink how offline demonstration data (including suboptimal demonstrations) can still be used as it appears uncommon in literature to do the type of offline buffer sampling our method (and RLPD, MoDem) do in on-policy algorithms.
>
> > Minor comment: " As a result, a curriculum for each demonstration is necessary as opposed to a curriculum constructed from all demonstrations as done in prior work in order to ensure noisy information arising from the multi-modality of demonstrations do not impact the reverse curriculum of each demonstration as much." -> Too long sentence.
>
> Revised!
>
> > You use \phi before you introduce it, and I didn't get what it's supposed to mean?
>
> We have revised the paper to clarify that $\phi$ is a hyperparameter indicating the ratio of the demonstration length to the episode horizon. A larger value would then mean shorter episodes relative to how long the demonstrator took, and vice versa.
>
> > "In this manner, initial states that sometimes receive return are prioritized the most, then initial states that receive no return, then initial states that are consistently receiving return." -> Not a very clear sentence, do you want to use 'than' instead?
>
> Thanks for pointing this out, we have made a revision to make it more clear.
>
> > Would be nice to discuss why the methods compared against were chosen out of all the possible RL algorithms out there (maybe in an appendix?) Would the others be unsuitable for the task (e.g. on-policy, not suitable for demonstrations etc.)
>
> This is a good point, we add to appendix B.5 how baselines were chosen and why some were not chosen. Generally we chose off-policy learning-from-demonstrations baselines as they are the most sample-efficient on our benchmarks and could not include some baselines because they do not have any code or tests on open-sourced tasks. We are actively trying to compare against at least one more baseline that uses state resets as we describe in response to reviewer gfCC.

---

### Official Review · Reviewer_UCiq · 2023-11-03

**Soundness:** 2 fair
**Presentation:** 4 excellent
**Contribution:** 2 fair
**Rating:** 3
**Confidence:** 4

**Summary:**

This paper deals with curriculum learning in cases where there are only a small number of demonstrations available. The proposed method specifically design the curriculum generation as two stages: one along the demonstration paths, another explore around demonstration and eventually cover the entire space. The experiments are performed on several learning from demonstration (LfD) benchmarks.

**Strengths:**

- The presentation of this paper is good.
- The experiment results are strong compared to multiple related baselines.

**Weaknesses:**

There are several issues/concerns on the method and experiments:

Method:

- The most important issue is the design of curriculum. Basically, the author tried to separate the curriculum of reset states into two stages: in the first one the reset states are along the demonstrated states, and in the second one the reset states gradually move away from the demonstrated states. This does not make sense. Why not just combine the two stages into one, i.e. a curriculum that includes explorations of reset/initial states along the demonstration and also away from demonstrated states? The reset states need to cover the entire space in the end anyway. I don't see any reason for a two-stage design to make sense. Conceptually, reverse curriculum and "forward curriculum" are almost the same thing, with the latter has a difference of weighting on the exploration area.

- What if there is no demonstration? Or intentionally forget about demonstration but just design curriculum that directly moves the reset/initial states away from goal? If this paper's assumption is that in the end, the initial/reset states should be able to cover the entire space anyway, then whether there is demonstrations provided should not matter: all starting states has to be explored sooner or later. I didn't see an ablation experiment to compare against the setting where no demonstration is available.

Experiment:

- As mentioned, it would be great to show the results under the setting where no demonstration is available, or the "forward curriculum"-only case. I understand the Maze task in Figure 6 tries to show this. But the experiment in Figure 6 is wrong: it does not show forward-curriculum-only is worse than reverse+forward curriculum. The reason it is worse is because forward-curriculum-only experiment mistakenly messed up with exploration (it should only explore in unexplored area, not around already explored demonstration area on blue lines). The explanation in the last two sentences of Section 5.2 is wrong too imo.

- In Figure 5, do all experiments have the same reset/initial state distribution (cover all possible states) at the end of their curricula? If so, it's so hard to understand why the more demonstration there is, the faster it can be trained.

- It seems that RFCL without "forward curriculum" works almost the same as RFCL with "forward curriculum" in most tasks (Fig 5). This again questions the necessity of having a second stage, as mentioned earlier. Without the two-stage setting, the novelty of this work is negatively impacted.

- I'm not sure if the tasks selected in this paper are suitable for the proposed method:
   - If the robot arms are moved by a positional controller, then the initial/reset position/states of the robot arm does not matter, because the positional controller should guide the robot arm/end effector converge to the goal position anyway regardless of initial states. Not much exploration is needed. So if this is the case, then the selected task cannot fully evaluate the potential of the proposed method.
   - If the robot arms are not moved by a positional controller, then why not do so?

**Questions:**

My questions are written in previous "Weakness" section. The authors can respond to the concerns/question written there.

---

> ### Author Response · Authors · 2023-11-12
> **Response to Reviewer UCiq [1/3]**
>
> Thank you for a comprehensive and detailed review, we welcome the feedback and suggestions and address the concerns below:
>
> > What if there is no demonstration? Or intentionally forget about demonstration but just design curriculum that directly moves the reset/initial states away from goal? If this paper's assumption is that in the end, the initial/reset states should be able to cover the entire space anyway, then whether there is demonstrations provided should not matter: all starting states has to be explored sooner or later. I didn't see an ablation experiment to compare against the setting where no demonstration is available.
>
> Thank you for raising this comment. We believe there may be a misunderstanding of the initial state distribution and what is inside it. In our paper, the initial state distribution $\mathcal{S}\_{\text{init}}$ refers to the distribution of states the environment resets to (e.g. reset robot to a rest position, place objects in random locations not near the goal locations). Critically, the **demonstrations provide states along a successful trajectory which are states outside of the initial state distribution**. Without a demonstration to reset states to, we have a very difficult exploration problem as the benchmarked tasks have high dimensional observation and action spaces, making it nearly impossible under sparse reward to explore to states near the goal and get reward.
>
> It is not possible to sample a state near the goal without using a demonstration, extra human-engineered code, or infeasible amounts of exploration. Designing a curriculum by hand would be time-consuming especially for these high dimensional tasks and human demonstrations (e.g. the demos in the Adroit benchmark) are easier to collect and need less expertise. That being said, we think it is interesting future work to explore how one can use simple human written curriculums (e.g. via language description) to accelerate policy learning.
>
> Running any RL algorithm without using demonstrations should achieve 0 success rate under the same compute budgets for the majority of tasks benchmarked due to the exploration problem under sparse reward. We will run no-demonstration experiments on some tasks with and without the forward curriculum and post results here.
>
> > The most important issue is the design of curriculum. Basically, the author tried to separate the curriculum of reset states into two stages: in the first one the reset states are along the demonstrated states, and in the second one the reset states gradually move away from the demonstrated states. This does not make sense. Why not just combine the two stages into one, i.e. a curriculum that includes explorations of reset/initial states along the demonstration and also away from demonstrated states? The reset states need to cover the entire space in the end anyway. I don't see any reason for a two-stage design to make sense. Conceptually, reverse curriculum and "forward curriculum" are almost the same thing, with the latter has a difference of weighting on the exploration area.
>
> Indeed, both curriculums are similar in that they control which states we reset to during training. We separate these into two curriculums as we want the policy to first prioritize learning to solve the task from the states in the demonstrations and then leverage the forward curriculum to prioritize other initial states sampled from the environment’s initial state distribution $\mathcal{S}_{\text{init}}$. By prioritizing demonstration states in a reverse curriculum first, the policy is more likely to collect useful data and reward first and learn to solve the task from a narrow subset of environment initial states. From there then the forward curriculum makes sense as the policy now actually can solve the task when starting from some initial states, it is up to the forward curriculum to help prioritize which initial states to train from.
>
> It is possible to merge the two into one curriculum where early during training we sample primarily along demonstration states and then over time sample more from the initial state distribution. For the purpose of explanation of the method it made more sense to give two separate names and in terms of practical implementation it works out better in code to have the two stage setup as demonstration states and states in $\mathcal{S}_{\text{init}}$ are mostly not similar (especially demo states near the goal). Furthermore, it’s unclear what the right distribution over demonstration states and initial states should be, while there are many options to try, we still achieved the strongest results via the two stage setup. We do believe there can be interesting future work to explore a more optimized, combined curriculum.

---

> ### Author Response · Authors · 2023-11-12
> **Response to Reviewer UCiq [2/3]**
>
> > As mentioned, it would be great to show the results under the setting where no demonstration is available, or the "forward curriculum"-only case. I understand the Maze task in Figure 6 tries to show this. But the experiment in Figure 6 is wrong…
>
> We agree the maze task could be more clear and clarify some details here in addition to revising the submission. The maze task is modified to mimic the robot tasks we benchmark by making the initial state distribution $\mathcal{S}\_{\text{init}}$ consist of any state not along the demonstration, and thus the chance of getting the sparse reward is difficult if you reset to initial states sampled from $\mathcal{S}\_{\text{init}}$. Moreover it is a **fully continuous state and action space, meaning the policy would still need to explore near states along the demonstration to perform well**. As a result, in the forward curriculum-only setting the policy can only reset to states not along the demonstration. It performs better than the no curriculum setting as the forward curriculum can prioritize sampling initial states close to the demonstration and is more likely to gather the necessary exploration data near the states along the demonstration. However, the forward curriculum-only setting performs less sample efficiently compared to the full RFCL method as the reverse curriculum enables the policy to collect necessary data around states along the demonstration and thus you see the fast improvement in success rate along the demonstration first in RFCL before the success rate improving on all states in $\mathcal{S}\_{\text{init}}$.
>
> > In Figure 5, do all experiments have the same reset/initial state distribution (cover all possible states) at the end of their curricula? If so, it's so hard to understand why the more demonstration there is, the faster it can be trained.
>
> After the reverse curriculum the policy can achieve high success rates if we initialize from the start of those demonstrations $\{s\_{i, 0}\}$, the first state of each demonstration $i$. Note that all $\{s\_{i, 0}\}$ are usually from $\mathcal{S}\_{\text{init}}$. With more demonstrations, the policy after reverse curriculum is more generalized across different initial states, meaning it can train faster as it does not need to spend as much time during the forward curriculum learning to generalize. The improved generalization can be seen in Fig. 5, the success rate curves are already trending up before the reverse curriculum completes (marked by the vertical dashed gray line) in higher-demonstration settings.
>
> Furthermore, more demonstrations means more ground-truth data of hard to reach states (e.g. peg inside the hole for the peg insertion task) with observation, action, and sparse reward labels to train with, which in general for any learning from demonstrations algorithm (e.g. DAPG, RLPD) will improve the performance.
>
> > It seems that RFCL without "forward curriculum" works almost the same as RFCL with "forward curriculum" in most tasks (Fig 5). This again questions the necessity of having a second stage, as mentioned earlier. Without the two-stage setting, the novelty of this work is negatively impacted.
>
> In particular, we emphasize that with a forward curriculum in RFCL we can achieve better demonstration efficiency. This is seen clearly by the blue line (RFCL) outperforming the green line (RFCL w/o forward curriculum) for settings with less demonstrations. For PegInsertionSide, with 10 demos or less the forward curriculum has much better success rates. For PickCube, the same occurs on the 1 demo ablation. When there are enough demonstrations, the reverse curriculum trained policy has much better generalization and thus the forward curriculum provides less noticeable (if any) benefits. The improved generalization of the reverse curriculum training is evidenced by the success rate curves already increasing before the dashed gray line in Fig. 5 (when the reverse curriculum completes).
>
> To further emphasize the importance of combining reverse and forward curricula, we will add some additional demo ablations comparing with and without forward curriculum results on some hard tasks (like Plug Charger) and post results here (and add to appendix due to space limitations).

---

> > ### Author Response · Authors · 2023-11-12
> > **Response to Reviewer UCiq [3/3]**
> >
> > > I'm not sure if the tasks selected in this paper are suitable for the proposed method: If the robot arms are moved by a positional controller, then the initial/reset position/states of the robot arm does not matter, because the positional controller should guide the robot arm/end effector converge to the goal position anyway regardless of initial states. Not much exploration is needed. So if this is the case, then the selected task cannot fully evaluate the potential of the proposed method.
> >
> > Thanks for asking about the controller, we revised appendix B.1 to include details on controllers. We use an end-effector delta pose or delta position controller (depending on what the benchmark provides), but most initial state randomness does not stem from the robot's joints being randomized. Regarding the appropriateness of task, there is a significant amount of exploration among initial states required as the initial state distribution contains randomizations of other properties of the tasks, such as the object shape in PegInsertionSide, object poses, and the goal locations (all tasks randomize goal, ManiSkill2 randomizes the most). For visuals of some of the initial state distribution, see Appendix B.2 / check the website section https://reverseforward-cl.github.io/#task-visuals-a.
> >
> > Robot arms are not moved by an absolute position controller (e.g. predicting absolute position, pose, or joint angles). While it could be, this is not the standard adopted in these benchmarks and used by other algorithms.
> >
> > We hope we are able to address all your concerns. Considering our response, would you be able to raise your score? We would be happy to discuss any further issues / questions you may have.

---

> > > ### Comment · Reviewer_UCiq · 2023-11-18
> > > **Reply**
> > >
> > > Thanks to the authors for the clarification and more information provided.
> > >
> > > Maybe I didn't make myself clear enough in my original review. But my main concern still remains unsolved while some more concerns came up.
> > >
> > > > It is not possible to sample a state near the goal without using a demonstration, extra human-engineered code, or infeasible amounts of exploration
> > >
> > > I disagree with this argument from the authors. If you want to define a valid task, you would need to define a set/distribution of goal states. If you cannot even sample a single valid goal state, how can you even define the set/distribution of goal states? I believe in practice, sampling a valid goal state would be much easier than defining the whole set/distribution of goal states. Or you should at least be able to sample *near* the goal states, as suggested by [A]. This argument (or assumption) by the authors is wrong imo.
> > >
> > > > We separate these into two curriculums as we want the policy to first prioritize learning to solve the task from the states in the demonstrations and then leverage the forward curriculum to prioritize other initial states sampled from the environment’s initial state distribution. By prioritizing demonstration states in a reverse curriculum first, ..., it is up to the forward curriculum to help prioritize which initial states to train from.
> > >
> > > For me, this reply from the authors raised even more concerns. I don't see why prioritizing demonstration states first and then other initial states is better than going in reverse direction from goal states to $S_\text{init}$ directly without considering the demonstration states.
> > >
> > > Another way to see this is to have an ablation experiment where you use **only the last states** in the demonstration trajectories, i.e. $\\{ s_{i,T_i} \\}$ in the authors' notation system, as the starting reset states of the generated reverse curriculum (if you feel you have no other ways of sampling goal states or near goal states). And then you move the reset states backward from $s_{i, T_i}$ to somewhere in $S_\text{init}$ without considering demonstration states $s_{i, 1}, s_{i, 2}, \ldots, s_{i,T_i-1}$. At the same time, you explore all other reset states including $S_\text{init}$ and the ones that reversely lead to $S_\text{init}$.
> > >
> > > I would imagine that this design would make more sense than moving backward along demonstrations (stage 1). From my experience, demonstrations are usually extremely noisy and dirty. It makes no sense to rely on the noisy data when you have other much better options.
> > >
> > > Sorry if I didn't make myself clear enough in my original review or I was asking too many additional experiments from the authors. I didn't imagine that the authors would say that "*It is not possible to sample a state near the goal...*", so I thought "*show the results under the setting where no demonstration is available*" is sufficient to describe this experiment.
> > >
> > > [A] Florensa, Carlos, et al. "Reverse curriculum generation for reinforcement learning." Conference on robot learning. PMLR, 2017.

---

> ### Author Response · Authors · 2023-11-18
> **Response by Authors**
>
> Thanks for the early reply and discussion!
>
> > I disagree with this argument from the authors. If you want to define a valid task, you would need to define a set/distribution of goal states. If you cannot even sample a single valid goal state, how can you even define the set/distribution of goal states? I believe in practice, sampling a valid goal state would be much easier than defining the whole set/distribution of goal states. Or you should at least be able to sample near the goal states, as suggested by [A]. This argument (or assumption) by the authors is wrong imo.
>
> To be clear, we refer to state as everything in the env including the robot state, object states etc. In practice, in all benchmarks we test on which are commonly used in the robot learning community, there are no environments that define a distribution of *useful* goal states. They always write simpler evaluation functions instead.
>
> One could spend human effort to reverse engineer the evaluation code to then write (near) goal state samplers, however for tasks like PickCube, you could easily sample states where the cube is at/near the goal, but the robot is not holding the cube. One could spend effort to write code to ensure that sampled goal states include the robot holding the cube, but this is exactly the goal of learning from demonstrations research: avoiding extra engineering by using demos (e.g. in the Adroit tasks we use human demos provided by the benchmark and perform well). This is definitely infeasible for tasks like the Adroit suite where there is a very high dimensional dextrous robot hand. In AdroitRelocate where the goal is to move the ball to the goal position, it is only feasible to sample goal states where the objects are at the goal but completely infeasible to sample valid robot hand states that would be holding the object at the goal due to the high dimensionality of the robot hand state (it has 30 degrees of freedom).
>
> We hope that it is clear that while it is possible to sample goal states, it's not easy to sample the useful goal states that include valid robot state that maintains the success. In AdroitRelocate, you can sample states where the ball is at the goal already and would get some success, but it becomes a difficult problem as the ball would instantly roll/fall away from the goal and now the algorithm has to learn from scratch how to grasp the ball and move it back to the goal making the task as difficult as it was before without the state reset. Hence, this is why demonstrations are critical, they describe a solution trajectory that is otherwise extremely difficult to sample with human engineered code as especially is the case in robotics tasks.
>
> > For me, this reply from the authors raised even more concerns. I don't see why prioritizing demonstration states first and then other initial states is better than going in reverse direction from goal states to $\mathcal{S}_{\text{init}}$ directly without considering the demonstration states...
>
> Unfortunately, it is a strong assumption made by [A] that you can move reset states backward from $s_{i,T_i}$ to somewhere in $S_{init}$ (in the case of [A] they assume you can sort of randomly sample actions to move from a goal state towards $S_\{\text{init}\}$). A simple counterexample to the claim that you can move reset states backward is the Adroit Door task. Many of the last states of demonstration trajectories in the Adroit suite end with the door open which yields success, but also end with the robot hand no longer grasping the door handle. This occurs on other tasks as well. Then to find states moving back to a state where the door is closed again is very difficult as you cannot easily sample states where a high dimensional robot hand is grasping the door handle correctly (this is the same problem in the forward direction). The reason why [A] was able to move reset states from the goal to the initial state distribution is because in their 2 robot tasks the object is always fixed to the robot (e.g. no grasping needs to be learned) which is not a realistic assumption of actual robotics. If the object wasn't fixed to the robot, many sampled reverse actions would lead to states where the robot isn't grasping the object and be hard to learn from.
>
> We are more than happy to run experiments for more convincing empirical evidence if necessary. We do note that the idea of trying to move a reset state backward to somewhere in $S_\{\text{init}\}$ by the method in [A] (taking reverse actions) is not possible in code for many tasks like Adroit, you can take a look at the Adroit code e.g. https://github.com/Farama-Foundation/Gymnasium-Robotics/blob/main/gymnasium_robotics/envs/adroit_hand/adroit_relocate.py and see that success is defined as the ball being at the goal, with no code about the hand grasping the ball (as this would be difficult).
>
> [A] Florensa, Carlos, et al. "Reverse curriculum generation for reinforcement learning." Conference on robot learning. PMLR, 2017.

---

> > ### Comment · Reviewer_UCiq · 2023-11-20
> > **Reply**
> >
> > Thanks to the authors for more comments.
> >
> > > We hope that it is clear that while it is possible to sample goal states, it's not easy to sample the useful goal states that include valid robot state …
> >
> > I am not saying that using demonstrations is not good. I am just saying that your argument “*It is not possible to sample a state near the goal without using a demonstration, extra human-engineered code, or infeasible amounts of exploration*” is wrong or too strong.
> >
> > Just like what I mentioned in my previous post: *if you feel you have no other ways of sampling goal states or near goal states, you can use the last states in the demonstration trajectories*.
> >
> > > Unfortunately, it is a strong assumption made by [A] that you can move reset states backward from $s_{i, T_i}$ to somewhere in $S_{init}$ …
> >
> > I don’t think this assumption is too strong.
> >
> > In fact, in your Forward Curriculum, you are making the exact same assumption, i.e. it is possible to move the reset states away from some demonstration states and eventually cover the entire $S_{init}$.
> >
> > > A simple counterexample to the claim that you can move reset states backward is the Adroit Door task. Many of the last states of demonstration trajectories in the Adroit suite end with the door open which yields success, but also end with the robot hand no longer grasping the door handle. This occurs on other tasks as well ...
> >
> > Sorry if I didn't make myself clear enough in my previous comment.
> >
> > I suggested $s_{i, T_i}$ (last states) because I thought $s_{i, T_i}$ is the only state in the trajectory $\tau_i$ that reaches the goal. From your description, it seems that there are many states in $\tau_i$ prior to $s_{i, T_i}$ that already reach the goal.
> >
> > If you think $s_{i, T_i}$ is not *useful* (e.g. due to the reason that the door is open but the robot hand is not grasping the door handle) and you don’t want it to be the reset state of the start of your reverse curriculum to $S_{init}$, you can possibly use an earlier demonstration state that is a useful goal state, e.g. $s_{i, T_i-10}$ where the door is open and the robot hand is still grasping the door handle, and then go from $s_{i, T_i-10}$ to $S_{init}$. These are just trivial engineering details.
> >
> > The fact that $s_{i, T_i}$ can potentially be a useless/invalid goal state (e.g. due to the reasons mentioned by the authors themselves) shows yet another weakness or mistake of this method. During the curriculum $\rho^r_{T_i}$, if $s_{i, T_i}$ is a useless/invalid goal state according to the authors, how can the learning in $\rho^r_{T_i}$ even be successful, considering $\rho^r_{T_i}$ only samples $s_{i, T_i}$ as the reset state? Are the reported experiment results even real, I wonder?
> >
> > Also, what would happen if $s_{i, T_i}$ is not the only useless goal state, but the last few demonstration states (e.g. from $s_{i, T_i-9}$ to $s_{i, T_i}$) are all useless goal states?

---

> ### Author Response · Authors · 2023-11-20
> **Response by Authors**
>
> Thank you for the additional discussion. We address them further below:
>
> > I am not saying that using demonstrations is not good. I am just saying that your argument “It is not possible to sample a state near the goal without using a demonstration, extra human-engineered code, or infeasible amounts of exploration” is wrong or too strong.
>
> This makes sense, apologies if it came off as too strong of a wording. Indeed there are some simpler tasks where it is possible to sample states along the goal (e.g. the tasks in [A] or a point maze). We just wish to emphasize this is not feasible for many tasks such as the Adroit suite or even the harder tasks in Metaworld and ManiSkill2 due to high state dimensionality of the robot and/or scene setup.
>
> > I don’t think this assumption is too strong. In fact, in your Forward Curriculum, you are making the exact same assumption...
>
> To clarify, the way [A] moves reset states backward from $s_\{i, T_i}$ to somewhere in $S_\{\text{init}\}$ is via taking reverse actions. Alternatively one could attempt to interpolate states but anything more advanced would likely be another (interesting) research direction. This is very different from the forward curriculum. The forward curriculum only performs some re-weighted sampling from $S_\{\text{init}\}$ whereas [A] needs to find a way to sample actions to find states between $S_\{\text{init}\}$ and $s_\{i, T_i}$, which is less than feasible for the Adroit set of tasks due to high dimensionality. We do not move states via e.g. adding noise to a state or taking actions like [A].
>
> Perhaps it might be clear if we define the different types of states in tasks as the following:
> 1. The initial states that an environment usually resets to (and one can sample from this freely)
> 2. The goal states where the environment would evaluate as success = True. This includes instances where in e.g. AdroitDoor the door is open (and the robot does not have to be currently grasping the door handle, and in code this is how it is checked for success). With some engineering you can sample goal states (that may not be easy to reverse from).
> 3. The states where typical success trajectories follow that help lead to goal states (but aren't in $S_\{\text{init}\}$ nor is a goal state). Many intermediate states in demos would fall in this category. We can label this "solution states" for now. These are generally not easy to sample without writing a whole solution to begin with.
> 4. All other states (e.g. ball in Adroit Relocate off the table)
>
> Then the method in [A] / idea of moving from goal states to initial states (whether the goal state is $s_{i, T_i}$ or $s_{i, T_i-10}$) is effectively looking to find solution states that are between initial states and goal states. E.g. a solution state in Adroit Door would be when the door is not completely open and the robot hand is still grasping the handle to open it. Another would be in a stack 3 blocks task where goal states are states with all 3 blocks stacked in a tower, if 2 / 3 blocks are already stacked and the robot is still in the progress of stacking the third block, this would be a solution state.
>
> We aren't sure how it would be possible in a task like stacking 3 blocks or adroit door to sample some of these solution states via the approach in [A]. In stacking 3 blocks, if you are given some goal state (say a useful one even) where the 3 blocks are stacked, and the robot hand is grasping the third block, to move back to a initial state without a demo you would need to sample reverse actions / do some extra engineering to solve effectively a "unstack 3 blocks" task which is just as hard as the original task. For Adroit Door finding solution states in reverse would be solving a "Close Door and lock it" task (the initial state of Adroit Door has the door locked via the handle). Demos overcome the need to have to find these solution states and avoid exploration problems.
>
> > The fact that $s_{i, T_i}$ can potentially be a useless/invalid goal state (e.g. due to the reasons mentioned by the authors themselves) shows yet another weakness...
>
> Sorry if we were unclear earlier around this. We are saying that $s_{i, T_i}$ is a goal state, but it may or may not be super useful (and engineering wise we always truncate demos until the last state that is a goal state by checking the environment evaluation). The reason RFCL can still work is because $s_{i, T_i}$ is still a goal state. Even though the robot hand in AdroitDoor is not grasping the door handle in $s_{i, T_i}$, the algorithm will take random actions and as long as it doesn't accidentally close the door the reverse curriculum will progress quite quickly until we reach the demonstration state where the robot hand is grasping the door handle and is already close to opening the door completely. The reverse curriculum progresses quickly because the useless goal states that sometimes appear in the end of demos all evaluate to success = True which progresses the curriculum.

---

> > ### Comment · Reviewer_UCiq · 2023-11-23
> >
> > Thanks to the authors for the comments and more information provided. I now have a better understanding of this method.
> >
> > I think the real problem with the proposed method is that, the authors (implicitly) made an assumption that $S_{init} \approx \\{ s_{i,0} \\}$, i.e. the task initial states $S_{init}$ is the same as the initial states of the demonstration trajectories $\\{ s_{i,0} \\}$ or covers roughly same small area in state space as $\\{ s_{i,0} \\}$. Though the authors did not explicitly mention this assumption in their problem setting or method description, this assumption can be inferred from their experiments and their previous comments to me.
> >
> > While it is OK to have a certain $S_{init}$ in the problem setting, I disagree with the assumption that $S_{init} \approx \\{ s_{i,0} \\}$ or $S_{init}$ being heavily biased towards $\\{ s_{i,0} \\}$. The initial states of the task don’t have to be constrained to the initial states of the demonstrations but can actually cover a much wider area of the states than what is present in the demonstrations.
> >
> > When you manually set the task initial states $S_{init}$ to be same or similar to demonstration initial states $\\{ s_{i,0} \\}$, of course your method will benefit from it because your method takes advantage of this manual bias of the task initial states. I don’t think your method can work well when the task initial states cover a much wider range that includes other valid states out of the scope of the demonstration states, due to the exact same reason you mentioned before with [A].
> >
> > Take the AdroitDoor as an example. It seems that all the task initial states are set to be the hand being fully stretched, placed at the front center of the door and facing downward. I would imagine the demonstrations have the same initial states. This does not make sense to me. Why can’t the task initial states be e.g. the hand being closed, placed on the right side of the door and facing to the left? I don’t think your method can work well with this initial state of the hands.
> >
> > In fact, I believe your method could have even more trouble than a method similar to [A], because when you biased the task initial states towards the demonstration initial states, your policy would also be heavily biased and would have trouble generalizing to other valid initial states. For example, the policy will only learn how to approach the door handle from the front side, but will have trouble generalizing to approaching from the right side.
> >
> > This is a common mistake by ML researchers. We have seen many such cases in the past where some ML researchers collected data with manually (and mistakenly) introduced bias and another group of ML researchers looked at the data and thought that the bias was universal and specifically proposed a problem formulation and/or method to take advantage of it. In your case, the bias is the distribution of task initial states.

---

> > ### Author Response · Authors · 2023-11-23
> > **Response by Authors**
> >
> > Thanks again for the active discussion, we are glad we are finding some common ground.
> >
> > We aren't sure if this was a point of confusion but we want to clarify that **we do not modify or manually bias the task initial states in any way during evaluation** (only during training via curriculum). All our results and figures are evaluating RFCL (and baselines) on task initial states that are not in or sometimes close to {$s_{i,0}$}. This is the case for tasks with large initial distributions like PegInsertionSide and even then RFCL can still solve those tasks.
> >
> > > I think the real problem with the proposed method is that, the authors (implicitly) made an assumption that...
> >
> > We agree we implicitly make an assumption however it is inaccurate to say $S_\{\text{init}\} \approx$ {$s_{i,0}$}. It would be more accurate to say all $s_{i, 0} \in S_\{\text{init}\}$ as we use very few demonstrations which do not approximate the initial state distribution that well for some harder tasks with wider initial state distributions. Your statement would be accurate for tasks like AdroitDoor for the reasons you point out. In all standard benchmarks we test on, the provided demonstrations always start from a state in $S_\{\text{init}\}$ due to how demos are collected by benchmark authors but this is the standard benchmark setting and all baselines make the same implicit assumption. We do not modify the initial state distributions of benchmark task when evaluating baselines or RFCL.
> >
> > Moreover, this is a limitation of all methods in this field when using sparse rewards and demos to solve hard tasks. If an algorithm is given too little demos, trying to solve from a completely out of distribution initial state will be difficult. We show this occurs for RFCL in the demo ablation (Fig. 5) where StackCube and PegInsertionSide cannot be solved easily when using too little demos. We attribute the failure to the inability of the neural network to generalize well to all initial states in $S_\{\text{init}\}$ as the reverse curriculum always completes, but in stage 2 training it fails to achieve good success on unseen initial states even if it trains on them (unless given sufficient demos as the demo ablation shows).
> >
> > > Take the AdroitDoor as an example. It seems that all the task initial states are set to be the hand being fully stretched, placed at the front center of the door and facing downward. I would imagine the demonstrations have the same initial states...
> >
> > We also do not understand why the authors of the Adroit suite made the task easier by making the initial state distribution small. We in fact clarify this is the reason why we originally ran demo ablations on ManiSkill2 as it has high initial state randomization in the start of Sec. 5.2. For example in ManiSkill2's PegInsertionSide, the peg's width, peg's length, peg's pose, and the box hole's pose are all randomized independent of each other. The randomizations are very large as shown in Fig. 13 / Appendix B.2. When given just 3 demos, we only see a small subset of initial state variations, but RFCL is still capable of sometimes solving the task (some seeds get near 100% while some get 0%) whereas all baselines are can't solve it. In PickCube we can solve the task completely from just a single demo, despite the task randomizing the pose of the cube. E.g. we can solve PickCube if the cube is on the left or right side of the robot despite seeing only one demo when the cube starts on the right.
> >
> > The important contribution of RFCL is it can reduce the number of demos needed to solve the task and with fewer demos {$s_{i, 0}$} more poorly approximates  $S_\{\text{init}\}$.
> >
> > While it is true that if our method is given demos with $s_{i, 0}$ not inside of or close to  $S_\{\text{init}\}$ it would fail, this is also true of all learning from demo algorithms. We do not claim that we can solve this particular problem tackling something very out-of-distribution. What is true however is that our method, despite being given very few demos that start from $S_\{\text{init}\}$, can still solve some complex tasks whereas baselines cannot.
> >
> > > In fact, I believe your method could have even more trouble than a method similar to [A], because when you biased the task initial states towards the demonstration initial states...
> >
> > We like to clarify that we do not bias task initial states toward demo initial states. We use the original initial state distributions from benchmarks. Although after reverse curriculum the policy is heavily biased and can only solve the task from the demo initial states, via the 2nd stage of training,the policy can begin to generalize to other valid initial states that it trains on as shown in the demo ablations and main results. This is accelerated by the forward curriculum as shown in ManiSkill2 (Fig. 5) and MetaWorld (Fig. 11).
> >
> > We hope these clarify your concerns and that you may consider raising your score. If not, we would be happy to hear additional feedback and concerns.

---

### Author Response · Authors · 2023-11-18
**Summary of revisions and new experiments based on reviewer feedback**

Summary of text revisions:
- Clarified what $\phi$ is, which is the demo length to episode horizon ratio (Sec. 4.1)
- Clarified the toy pointmaze has continuous state/action space (Sec. 5.2)
- Clarified the default controller type used for benchmarked tasks (Appendix B.1)
- Clarified how baselines were selected (Appendix B.5)

We also ran some additional experiments based on feedback from reviewers and have included some of them into the paper:

Reviewer UCiq raises a good point about testing on a forward curriculum only and no demo setting. We detail the results below:

Forward Curriculum Only setting: We ran this on our demo ablation study on ManiSkill2 tasks and the results show that without the reverse curriculum, it does only slightly better than RLPD but otherwise needs many demos to succeed. Results are updated on Figure 5 (new orange line).

Without demos setting: Without any demos, regardless of whether a forward curriculum is used or not, gets 0 success rate on ManiSkill2 tasks after 2 million samples.

To also more strongly show the contributions of the forward curriculum, we ran additional
RFCL vs Reverse Curriculum Only (RFCL w/o forward curriculum) experiments on the Metaworld suite and show a demo ablation in Appendix A.3. They show that when given very few demonstrations, the forward curriculum helps accelerate learning significantly, whereas when given a sufficient number of demonstrations, the impact of the forward curriculum is minimal. The conclusions are the same as those drawn from the original demo ablation experiments in Figure 5.

Reviewer gfCC asks a good question about comparisons with baselines that use state reset. One baseline is the method proposed by Nair et. al. [1] which uses uniform state reset. We originally compared our state reset approach against [1] and results in Table 1 show that their approach of state reset (labeled as uniform) fails to reliably solve all ManiSkill2 tasks in the allotted time budget of 1 million samples. Table 1 shows results using the same core RL algorithm used in RFCL but varying the way state reset is used.

We further ran new experiments comparing against [1] on their difficult Stack 3 environment with sparse rewards. In appendix A.4 we experiment with RFCL on the stack 3 task and show that our success rate after 10 million steps when using just 20 demonstrations outperforms [1] significantly as [1] has a lower success rate even after 100 million steps while using 100 demonstrations. Similarly, RFCL has better sample efficiency and demo efficiency on a harder version of their Pick and Place environment (ManiSkill2’s Pick Cube task) as detailed in our response to reviewer gfCC.

We hope the clarifications and additional experiments address each reviewer's concerns. We thank all the reviewers again for their efforts in reviewing and helping us improve the submission! Please let us know if we can make any additional clarifications or experiments to improve the review scores.

---

### Comment · Area_Chair_RCMC · 2023-11-23
**Author-Reviewer discussion period ending *very* soon**

Thank you to the reviewers for responding. The authors have put great effort into their response, so can I please urge reviewer Rzng to answer the rebuttal.
Thank you!

---

> ### Author Response · Authors · 2023-11-23
> **Comment by Authors**
>
> Thanks for putting out the reminder. In fact all reviewers have responded, Rzng just edited their original review instead with a “post rebuttal” part.
>
> We are happy to see all reviewers partake in discussion. Even if we disagree on certain points the discussion was pleasantly precise and great to partake in.

---

> > ### Comment · Area_Chair_RCMC · 2023-11-23
> >
> > Acknowledged; thank you reviewer Rzng

---

### Meta-Review · Area_Chair_RCMC · 2023-12-06

**Metareview:**

The paper introduces RFCL, a novel approach aiming to enhance reinforcement learning (RL) with offline demonstration data to address the challenge of sparse reward-based complex task solving. RFCL comprises a dual-stage curriculum learning strategy, uniquely leveraging multiple demonstrations through a per-demonstration reverse curriculum generated via state resets. Initially, a reverse curriculum yields an adept policy focused on a narrow state distribution, aiding in mitigating exploration challenges. Subsequently, a forward curriculum accelerates training, refining the initial policy to encompass the entire initial state distribution, thereby enhancing both demonstration and sample efficiency. The paper demonstrates how this combined approach significantly improves efficiency compared to existing learning-from-demonstration baselines.

Reviewers applaud the paper's clear presentation and strong empirical results against various related baselines. However, concerns surfaced regarding both methodological aspects and experimental execution. Reviewer UCiq questions the rationale behind the two-stage design of the curriculum, suggesting the conceptual similarity between reverse and forward curricula and urging for comparisons with settings having no demonstrations. Experiment criticisms include flawed executions in the Maze task and doubts about the suitability of the selected tasks to fully evaluate the proposed method's potential. Reviewer Rzng appreciates the paper's demonstrated performance but advises improving the algorithmic summary for better clarity, critiquing specific numerical choices in the method. Reviewer gfCC acknowledges the approach's novelty but raises concerns about its applicability in real-world scenarios, perceiving similarities to other methodologies and suggesting better comparisons with non-curriculum-based algorithms.

Post rebuttal. there is no clear consensus among reviewers on whether to accept --- with 1 reviewer suggesting accept, another suggesting reject, and 1 borderline. During the rebuttal period, there was a lengthy discussion between authors and reviewer UCiq (which lowered their score from 5 to 3 post rebuttal). This AC believes the main reason reviewer UCiq lowered their score was because the reviewer realised that:

```...the task initial states is the same as the initial states of the demonstration trajectories or covers roughly same small area in state space...```

and asserted that:

```The initial states of the task don’t have to be constrained to the initial states of the demonstrations but can actually cover a much wider area of the states than what is present in the demonstrations.```

This AC believes that the paper evaluated on a set of fairly standard offline/online RL/LfD benchmarks, and having a limited set of initial start states is standard benchmark setting and all baselines make the same implicit assumption.

**Justification For Why Not Higher Score:**

There was not a strong push to accept from the majority of reviewers; and string assumptions about resetting simulator state which severally limits its usefulness.

**Justification For Why Not Lower Score:**

N/A

---

### Decision · Program_Chairs · 2024-01-16

Accept (poster)